# DisA Restrains the Processing and Cleavage of Reversed Replication Forks by the RuvAB-RecU Resolvasome

**DOI:** 10.3390/ijms222111323

**Published:** 2021-10-20

**Authors:** Carolina Gándara, Rubén Torres, Begoña Carrasco, Silvia Ayora, Juan C. Alonso

**Affiliations:** Department of Microbial Biotechnology, Centro Nacional de Biotecnología, CNB-CSIC, 3 Darwin St, 28049 Madrid, Spain; carolina.rosa@gmail.com (C.G.); rtorres@cnb.csic.es (R.T.); bcarrasc@cnb.csic.es (B.C.)

**Keywords:** replication stress, DNA damage signal, fork reversal, c-di-AMP, RuvAB, RecU, DisA

## Abstract

DNA lesions that impede fork progression cause replisome stalling and threaten genome stability. *Bacillus subtilis* RecA, at a lesion-containing gap, interacts with and facilitates DisA pausing at these branched intermediates. Paused DisA suppresses its synthesis of the essential c-di-AMP messenger. The RuvAB-RecU resolvasome branch migrates and resolves formed Holliday junctions (HJ). We show that DisA prevents DNA degradation. DisA, which interacts with RuvB, binds branched structures, and reduces the RuvAB DNA-dependent ATPase activity. DisA pre-bound to HJ DNA limits RuvAB and RecU activities, but such inhibition does not occur if the RuvAB- or RecU-HJ DNA complexes are pre-formed. RuvAB or RecU pre-bound to HJ DNA strongly inhibits DisA-mediated synthesis of c-di-AMP, and indirectly blocks cell proliferation. We propose that DisA limits RuvAB-mediated fork remodeling and RecU-mediated HJ cleavage to provide time for damage removal and replication restart in order to preserve genome integrity.

## 1. Introduction

In living cells, replication fork progression is frequently hindered by obstacles in and on the DNA template [1,2,3,4]. Cells may use several strategies when DNA replication is challenged by this stress: replication forks stall, DNA polymerases uncouple, lesions can be simply skipped by the replisome forming single-stranded DNA (ssDNA) gaps, or the stalled fork is pushed backwards to convert it into a Holliday junction (HJ)-like structure by allowing the pairing of the two nascent strands and rewinding of the parental strands (fork reversal, known also as fork regression) [5,6,7,8,9]. Nevertheless, in *Escherichia coli*, fork reversal appears to be a less relevant response to DNA damage, and is infrequent in wild-type (*wt*) cells, because reversed forks are susceptible to nucleolytic degradation of the regressed nascent DNA arms. In fact, RecBCD (counterpart of *Bacillus subtilis* AddAB) prevents or removes reversed fork structures, and in the Δ*recBCD* context, the reversed forks are processed by the RuvAB translocase and cleaved by the RuvC (counterpart of *B. subtilis* RecU) HJ resolvase, leading to fork breakage and one-ended double-strand breaks (DSBs) [7,8,10]. In sharp contrast, in mammalian cells [11] or during the early stage of *B. subtilis* spore revival [12,13], dedicated mechanisms are actively involved in the formation and integrity of reversed forks. Therefore, the analysis of the repair functions active in reviving *B. subtilis* spores gives clues of the proteins that contribute to genome integrity after fork stalling, because breakage of a reversed fork should be pathological during phases where only one genome copy is available, and cells should prevent it (unless stated otherwise, indicated genes and products are of *B. subtilis* origin).

When the DNA of an inert mature haploid spore is damaged by ionizing radiation, and then the spores are synchronously revived under unperturbed conditions, spores lacking both AddAB and RecJ exonucleases are as capable of repairing the damage as the *wt* control [12], showing that long-range end-resection functions play a minor role in removing a replicative stress. Indeed, these proteins are synthesized after initiation of spore replication and prior to cell elongation [14,15]. By contrast, spores required the recombinase RecA, its mediators and modulators, the branch migration translocases (RuvAB, RecG), the HJ resolvase RecU, and the DNA damage checkpoint sensor DisA for replication fork rescue [12,13]. We can envision that during the first replication cycle of those reviving spores, in the absence of an intact homologous template and end-resection functions, the fork may be reversed to remove a replicative stress. This way, the lesion is placed on a double-stranded (ds) configuration to permit its removal by means of base- and nucleotide-excision repair. Then, the regressed nascent complementary DNA arms can potentially allow for the bypass of template DNA obstructions by error-prone and error-free DNA damage tolerance (DDT) sub-pathways that contribute to bypass or circumvent the lesion. Nevertheless, fork remodeling could be controlled by a poorly characterized function to avoid the formation of highly detrimental DSBs.

In vivo analyses showed that DisA is a DNA sensor protein found in the Firmicutes and Actinobacteria phyla, among others [16]. DisA forms a focus that moves rapidly along the chromosome scanning for “perturbations” but pauses its movement upon DNA damage and delays entry into sporulation until the offending lesion is removed [17]. DisA still pauses in the absence of long-range end resection (∆*addAB* ∆*recJ* cells) [18], but it does not pause in the absence of RecO and RecA. This suggests that DisA pauses at a DNA structure formed after RecO and RecA engagement at a site of replicative stress, so that the primary signal recognized by DisA is a lesion-containing gap or a branched intermediate rather than DNA ends or DSBs. While scanning the chromosome, DisA converts a pair of ATPs into the essential second messenger cyclic 3′, 5′-diadenosine monophosphate (c-di-AMP) [19]. Paused DisA, however, drops the synthesis of c-di-AMP in response to replication perturbations, as those induced by methyl methane sulfonate (MMS) [20], to levels comparable to that in the absence of DisA [21]. DisA is composed of an N-terminal globular domain featuring diadenylate cyclase (DAC) activity, and a C-terminal RuvA-like HJ DNA-binding domain, separated by a central helical domain [22]. Octameric DisA, upon binding to a branched DNA structure, undergoes a conformational change that inhibits its c-di-AMP synthesis [22,23,24]. Low c-di-AMP levels indirectly increase (p)ppGpp synthesis [25], and (p)ppGpp blocks DNA replication by inhibiting DNA primase activity [26]. This DisA fail-safe mechanism of coordinating repair-by-recombination and replication is indirect, because DisA neither compromised PriA-dependent initiation of DNA replication nor affected replication elongation using an in vitro reconstituted replication assay [13].

DisA acts selectively at stalled forks, since Δ*disA* cells show a similar resistance to that of *wt* cells upon exposure to H_2_O_2_ or nalidixic acid, which induce single-strand nicks or DSBs, respectively [21,23]. Single-molecule studies revealed that DisA shows a dynamic movement in exponentially growing *wt* cells, but it becomes static when branched recombination intermediates accumulate, as in the absence of the RecU HJ resolvase or the RecG branch migration translocase [23]. The current model for DisA action is that this sensor protein may control the response to a replicative stress until the damage is removed, and balances the benefits and risks of fork remodeling [13,17,18,27]. It is poorly understood whether functions that remodel branched DNA structures affect c-di-AMP synthesis.

One may question which function(s) remodel a stalled replication fork in *B. subtilis*. In *E. coli*, stalled forks are processed by the RecA, RecG, RuvAB, or RecQ remodelers [28,29,30,31,32] or the ssDNA gap left by the skipped lesion is extended by RecJ and RecQ to facilitate RecA-mediated gap repair [7,28,29,30,31,32]. Among the functions required for *B. subtilis* spore survival upon DNA damage are RecA, RuvAB, RecU, RecG, and DisA, and genetic data have shown that the *disA* gene is epistatic to *recA*, *ruvA*, *ruvB*, *recU*, and *recG* genes upon exposure to ionizing radiation [12,13]. This suggests that DisA acts with these proteins in common mechanisms to ensure the stability of the stalled forks and the maintenance of cell survival. Previous studies have explored how DisA could modulate RecA or RecG activities [18,27,33], but the interplay between DisA and the resolvasome (RuvAB-RecU) is poorly understood.

When a replicative stress occurs, the RuvAB levels increase as part of the SOS response, and the DisA and RecU pools increase as part of the cell envelope stress response [34,35,36], suggesting a temporal link between increasing levels of these proteins and a replication perturbation. RuvAB and RecU are the *B. subtilis* counterpart of the *E. coli* RuvABC (RuvABC*_Eco_*) resolvasome [37,38,39,40,41]. RuvA and RuvB are among the most ubiquitous bacterial proteins, whereas RecU is selectively found in bacteria of the Firmicutes and Tenericutes phyla. The RecU structure, which is unrelated to RuvC, shares homology to certain bacteriophages and archaeal HJ-resolving enzymes [42,43]. The resolvasome might act at reversed forks (HJs) in response to a stalled fork and is crucial for the processing of double HJs during canonical DSB repair [5,6,7]. RuvA specifically binds and stabilizes HJs [37]. A RuvA-HJ complex is the first crucial step for RuvB loading and for the formation of an ATP-dependent RuvAB motor [44,45]. RuvB interacts with RecU [36]. RecU specifically binds HJ DNA [46,47]. Finally, once RuvAB-mediated branch migration exposes the RecU cognate site, RecU cleaves the HJ to yield two nicked duplexes [46,48].

Taking the in vivo data into account, in this study, we biochemically explored how DisA could modulate the stability of DNA structures that mimic a stalled or reversed replication fork, by analyzing its interplay with RuvAB and RecU. We show that DisA contributes to reducing chromosome degradation. DisA, which binds HJ DNA with high affinity in the presence of physiological Mg^2+^ concentrations, physically interacts with RuvB. RuvAB branch migrates a fixed (HJ-J3) or mobile (HJ-J4) DNA to restore a replication fork, but it poorly converts a stalled fork into an HJ-like structure. DisA inhibits RuvAB-mediated ATP hydrolysis and processing of HJs. DisA, which does not interact with RecU, inhibits RecU-mediated resolution of HJs. In the presence of RuvAB or RecU bound to HJ-J3 DNA, DisA-mediated c-di-AMP synthesis is strongly inhibited. These data suggest that DisA may ensure fork stability by timely coordinating RuvAB- and RecU-mediated processing of branched intermediates at the damaged replication fork.

## 2. Results

### 2.1. DisA Preferentially Binds DNA at High Mg^2+^ Concentrations

Single-molecule studies revealed that: (i) the dynamic movement of DisA pauses at a RecA-bound lesion-containing gap in sporulating cells [18]; (ii) DisA becomes static in the Δ*recU* context, where branched intermediates accumulate [23]; and (iii) DisA-mediated c-di-AMP synthesis is unaffected by duplex DNA but is inhibited by branched intermediates [22]. Thus, it is likely that octameric DisA, which consists of two peripheral quartets of helix-hairpin helix (HhH) domains connected to the central DAC domains by a helical spine [22], preferentially binds to stalled or reversed forks in a universe of duplex chromosomal DNA through its HhH domains that resemble that of RuvA. To characterize the specific DNA structure recognized by DisA, DNA binding was analyzed by electrophoretic mobility shift assays (EMSAs). The addition of 0.2% glutaraldehyde prior to separating the DisA-DNA complexes, however, was necessary to detect and visualize them.

In the presence of physiological Mg^2+^ concentrations, DisA bound both dsDNA and HJ-J3 DNA with similar apparent DNA binding constant (K_Dapp_) values (2.6 ± 0.3 nM and 3.0 ± 0.5 nM, respectively), but the complexes formed with these two substrates were different (Figure 1A,B). A fast-moving complex was observed with dsDNA, whereas DisA-HJ-J3 complexes tend to remain trapped in the well even at low acrylamide concentrations, suggesting that the RuvA-like HhH tetrameric domains arranged at each side of the dumbbell-shaped DisA octameric structure interact differently with duplex and HJ-J3 DNA [22]. DisA also bound flayed DNA (unreplicated fork DNA) and single-stranded (ss) DNA with a similar high affinity (K_Dapp_ 3.2 ± 0.2 nM and 3.3 ± 0.1 nM, respectively) (Appendix A Appendix A), and formed complexes that remained trapped in the well (data not shown). This unexpected result suggests that (an) auxiliary protein(s) or cofactor specifically recruits DisA to the stalled fork, or HJ DNA. Alternatively, we did not use the proper conditions for DisA.

To test the latter hypothesis, we searched for the optimal conditions for the DAC activity of DisA. DisA-mediated c-di-AMP synthesis was not detected (*p* < 0.01) in the absence of Mg^2+^ ions (presence of 5 mM EDTA). In the presence of 1 mM MgCl_2_, DisA poorly catalyzed c-di-AMP synthesis, whereas the DAC activity of DisA was significantly increased in the presence of 10 mM MgCl_2_ (Appendix A). In the presence of 10 mM MgCl_2_, DisA-mediated c-di-AMP synthesis was inhibited by the addition of HJ-J3 DNA (*p* < 0.01) but not by dsDNA (*p* > 0.1) (Appendix A) [22]. Similarly, binding to DNA was best at high than at low MgCl_2_ concentrations or in the presence of EDTA (*Annex 1*, Appendix A). It is likely that in the presence of physiological Mg^2+^ concentrations (10 mM), DisA forms a stable macro-complex with HJ-J3 DNA that is not entering in a gel and suppresses its DAC activity. DisA, however, forms only a transient interaction with duplex DNA, which was captured in our EMSA assays because of the use of glutaraldehyde. Thus, the presence of dsDNA does not affect DisA-mediated DAC activity (Appendix A).

Finally, the effect of nucleotide cofactors in HJ-J3 binding was tested. DisA, in the ATP-bound form, bound HJ-J3 DNA with slightly less affinity (K_Dapp_ 5.5 nM ± 0.3 nM) (Figure 1C). Saturating c-di-AMP concentrations further reduced the interaction of DisA with HJ-J3 DNA (Figure 1C), suggesting that when DisA converts ATP into c-di-AMP, this interferes with HJ DNA binding. It is likely that: (i) DisA interacts transiently and dynamically with duplex chromosomal DNA, and such interaction is not sufficient to suppress DisA-mediated c-di-AMP synthesis (Figure 1A and Appendix A); (ii) DisA bound to branched intermediates (e.g., a stalled or reversed fork) forms a large molecular mass complex that undergoes a conformational change to suppress DisA-mediated c-di-AMP synthesis (Figure 1B and Appendix A); and (iii) (an) unknown protein(s) might contribute to recruit and stabilize DisA at branched intermediates, with RecA, which physically interacts with DisA [18], being a good candidate to help DisA loading.

### 2.2. DisA Interacts with RuvB

A genetic interaction of DisA with the branch migration translocases (RuvAB or RecG) in response to DNA damage was inferred from survival assays [13]. The RecG enzyme fails to form stable complexes with DisA [27]. Here, we analyzed if RuvAB interacts with DisA. To test this, a bacterial two-hybrid system was used (see the material and methods) (Figure 2A–C).

The coding sequence of the genes was fused to the 5′ or 3′ sequence of either the T18 or T25 region of the *Bordetella* adenylate cyclase gene, as described [18]. Using this system, we confirmed that DisA, RuvA, or RuvB interact with themselves (Figure 2A–C), because such interaction activates the cAMP-bound catabolite activator protein to induce the expression of β-galactosidase. This leads to the appearance of blue-colored colonies, due to the breakdown of X-gal in the medium, as observed for the Zip control (Figure 2A–C). We observed that the interaction of DisA with RuvB induces β-galactosidase expression (Figure 2A), whereas a physical interaction of DisA with RuvA was not deduced (Figure 2C).

The DisA HhH domains structurally resemble that of RuvA [22]. To evaluate whether DisA interacts with RuvB through its C-terminal HhH RuvA-like DNA-binding domain, the *disA*ΔC290 mutant gene [18] was fused to the T18 or T25 regions. The interaction of RuvB with DisA ΔC290 variant induced β-galactosidase expression to levels comparable to that of the Zip control (Figure 2B). It is likely that the DNA-binding domain of DisA is dispensable for its interaction with RuvB.

### 2.3. DisA Coexists with RuvAB on HJ DNA

To test whether DisA works in concert with RuvAB, the RuvA and RuvB proteins were purified and EMSAs were performed. The RuvB–DisA interaction could not be studied by EMSA because RuvB as RuvB*_Eco_* fails to form a stable complex with HJ DNA, even in the presence of the non-hydrolysable ATP analogue ATPγS and of 0.2% glutaraldehyde addition to fix any pre-existing RuvB-DNA complex (data not shown). Similar results were previously reported [37,41]. RuvAB forms a slow-moving complex with HJ DNA that is trapped in the well [41]. Therefore, the existence of a hypothetical DisA-HJ DNA-RuvAB complex could not be distinguished from the DisA-HJ DNA complex using EMSA (see Figure 1B). The first step in HJ resolution by the RuvAB-RecU resolvasome is the formation of the RuvA-HJ DNA complex [41]. Thus, the formation of putative RuvA-HJ DNA-DisA complexes was analyzed by EMSA.

The RuvA protein binds HJ-J3 DNA preferentially in the presence of EDTA and binding is strongly reduced in the presence of 10 mM MgCl_2_ [49,50,51]; therefore, the experiments were performed in the presence of 1 mM MgCl_2_, although this is not the optimal condition for DisA binding (see Appendix A and Appendix A). RuvA bound [γ^32^P]-HJ-J3 DNA with a K_Dapp_ of 10 ± 2 nM, and saturating RuvA concentrations led to the formation of a slow-moving R-II complex (Figure 2D, lanes 2 and 3), which corresponds to two RuvA tetramers bound to the same HJ molecule, as observed previously in the presence of EDTA [50]. DisA bound [γ^32^P]-HJ-J3 DNA and formed large molecular mass complexes that mainly remained trapped in the well (Figure 2D, lane 5), as the ones observed at high Mg^2+^ concentrations (Figure 1B). DisA seemed to facilitate RuvA-HJ DNA complex formation (Figure 2D, lane 6 vs. 2), and a new RuvA-HJ-J3 DNA-DisA complex (R+A) was observed (Figure 2D, lanes 7 and 9). It is likely that DisA and RuvA bound HJ DNA with a certain degree of cooperativity, and that both proteins may co-exist in the same HJ DNA molecule.

To address whether DisA and RuvAB co-localize and interact with the HJ, a DNase I footprint assay was performed. The protein-[γ^32^P]-HJ-J3 DNA complexes were pre-formed (5 min at 37 °C), and then fixed DNase I, 5 mM ATPγS and MgCl_2_ up to 10 mM were added, because DNase I requires 10 mM MgCl_2_ for its catalytic activity (Figure 2E). The junction or cross-over region of the HJ DNA is single stranded, and thus insensitive to DNase I attack (Figure 2E, *junction*). RuvB·ATPγS did not protect the HJ DNA from DNase I attack, confirming that RuvB needs to be recruited on the HJ DNA (Figure 2E, lane 2). RuvA, bound at the junction, protected both arms of the labeled HJ strand, with a footprint of ~26-nt, and revealed some hypersensitive sites (Figure 2E, lanes 3 and 4, frame and asterisks). RuvA even protected the DNA substrate from the observed spontaneous cleavage (see lane C, control). In the presence of stoichiometric RuvB·ATPγS and RuvA, HJ-J3 DNA was protected from DNase I attack, with an extended footprint of ~40-nt (Figure 2E, framed lanes 5 and 6). This extended footprint was also observed for the RuvAB*_Eco_* complex [52].

DisA binding showed an extended region of protection from DNase I attack (40–45 bp DNA) (Figure 2E, lanes 7 and 8) [27]. This is consistent with the hypothesis that the DisA HhH domains interact with the different arms of the HJ, forming large complexes as observed above. These results indirectly suggested that DisA could prevent degradation of the extruded nascent strands of a reversed fork. At present, we cannot rule out that more than one DisA octamer is recruited to the HJ DNA.

When DisA was added to pre-formed RuvAB-HJ DNA complexes or vice versa, at a DisA:RuvAB molar ratio of 1.5:1 to 3:1, an extended region of protection from DNase I attack, similar to those for DisA, was observed (Figure 2E, lanes 5–8 vs. 9–16). The footprint also showed the patterns observed for RuvAB, as the specific hypersensitive bands (denoted by asterisks), suggesting its presence on the HJ DNA together with DisA. Since the stoichiometry of the observed signals is similar in the different tracks, we are confident that both DisA and RuvAB are bound to the same molecules of HJ DNA.

### 2.4. DisA May Protect Cells from Chromosome Breaks

DNase I footprinting studies suggested that DisA might shield branched intermediates so that it could have protecting activity in vivo, thus avoiding the degradation of the reversed fork. We then tested whether DisA protects DNA from degradation during rapid growth in LB medium in unperturbed conditions or upon MMS-induced replicative stress, quantifying the accumulation of chromosomal fragments in *wt* and Δ*disA* cells by pulsed-field gel electrophoresis (PFGE) (Figure 2F). It is worth noting that circular DNA and chromosome-sized branched replication intermediates are unable to migrate into pulsed-field gels and remain in the well [20,53].

In the absence of DNA damage, a modest but significative increase in the amount of degraded DNA was observed in Δ*disA* cells when compared to the *wt* control (*p* < 0.01) (Figure 2F,G). To test whether this degradation is due to RuvAB-RecU-mediated cleavage of the reversed fork as observed in *E. coli* [10] or during canonical DSB repair [37], we tested chromosomal degradation in the Δ*recU* context (Appendix A). In the absence of the RecU HJ resolvase, no significative decrease in the amount of degraded DNA was observed (*p* > 0.5) (Appendix A). It is likely that in the absence of DisA, prolonged replication stress may cause replication fork collapse and degradation but not in the absence of RecU.

Then, DNA replication was perturbed by the addition of MMS. MMS, which mainly methylates purines, causes replication fork slowing and fork stalling [20]. In the presence of 10 mM MMS, which reduces ~3-fold the survival of *wt* cells [23], an increase in DNA fragmentation was observed when compared to the absence of MMS (*p* < 0.05) (Figure 2F). The quantitative analysis of three independent experiments also revealed the increase of chromosomal fragmentation in Δ*disA* cells upon treatment with MMS. The levels of chromosomal degradation in the Δ*disA* context were higher than in the *wt* control (Figure 2G). These results suggest that: (i) DisA may contribute to protect paused or stalled forks from degradation in unperturbed replicating cells, and (ii) MMS induces chromosomal fragmentation in both *wt* and Δ*disA* cells.

### 2.5. DisA Cannot Activate RuvAB to Catalyze Fork Reversal

In *E. coli*, HJ intermediates accumulate in Δ*ruvAB* cells but not in the Δ*ruvAB* Δ*recG* context [54], suggesting that RuvAB*_Eco_* by itself may not reverse a stalled fork. To test whether RuvAB converts a stalled fork with a gap in the leading or in the lagging strand onto a reversed fork, and if DisA affects the rate of such conversion, two synthetic stalled replication forks, with a 15-nt ssDNA gap either on the leading strand (forked-Lead) or on the lagging strand (forked-Lag), were used. These substrates have heterologous arms to prevent spontaneous branch migration; thus, if RuvAB mediates fork reversal, the resulting product should migrate as a flayed DNA (unreplicated fork) (Figure 3).

In the presence of ATP, RuvAB (15 nM) did not unwind forked-Lead or forked-Lag DNA, and a flayed intermediate that would be the product of the backwards pushing of the replicated fork to displace both nascent strands was not observed (Figure 3A,B, lane 6). In contrast, when the branch migrating translocase RecG (15 nM) was incubated with these forked-Lead or forked-Lag DNA substrates and ATP, the flayed DNA product was observed [27].

To test whether DisA changes this outcome by its interaction with RuvB, increasing DisA concentrations were added to the reactions with the forked-Lead or forked-Lag DNA substrates. As expected, DisA does not show any activity over these substrates (Figure 3A,B, lanes 3–5). Independently of the order of protein addition, DisA did not facilitate RuvAB-mediated formation of reversed forks by branch migrating these DNA structures (Figure 3A,B, lanes 7–12 vs. 6). Similarly, stalled forks are poor substrates for RuvAB*_Eco_ in vitro* [31].

### 2.6. DisA Bound to HJ DNA Inhibits RuvAB-Mediated Fork Restoration

In vitro, both RuvAB*_Eco_* and RuvAB efficiently branch migrate an HJ-like structure to restore a replication fork [30,31,41]. We showed that in the presence of ATP, RuvAB (15 nM), bound to a HJ (HJ-J4 DNA), produced the flayed duplex product, but it failed to further unwind the flayed duplex intermediate to render an ssDNA substrate (Figure 3C, lane 5 vs. 12). Similar results were observed when RuvAB was incubated with HJ DNA that cannot spontaneously branch migrate (HJ-J3 DNA) [41]. DisA did not unwind the HJ-J4 DNA (Figure 3C, lanes 2–4).

We tested whether DisA affects RuvAB-mediated fork restoration. DisA pre-incubated with HJ-J4 DNA significantly inhibited RuvAB-mediated branch migration (*p* < 0.01) even at a DisA:RuvAB molar ratio of 0.8:1 (Figure 3C, lane 9 and 3D). When DisA was added to the pre-formed RuvAB-HJ-J4 DNA complexes, however, no inhibition was observed at a DisA:RuvAB molar ratio of 0.8:1, but at a 1.6:1 DisA:RuvAB molar ratio, ~50% of the HJ-J4 DNA was not unwound (Figure 3C, lanes 6 and 7 and 3D). In the presence of an excess of DisA (DisA:RuvAB molar ratio of 3:1), RuvAB-mediated branch migration was significantly inhibited (*p* < 0.01) (Figure 3C, lane 11 and 3D). The different outcome observed by altering the order of protein addition reveals a genuine activity associated with DisA rather than any inhibitory component present in the protein preparation (e.g., a protease). We envision that DisA bound to the HJ DNA blocks branch migration. Alternatively, DisA acting as a potent roadblock inhibits any translocase, including RuvAB. Nevertheless, we consider this hypothesis unlikely, because the replicative accessory DNA helicase PcrA (a member of the conserved UvrD-FBH1 family of 3‘→5’DNA helicases) efficiently processes a tailed HJ-DNA even in the presence of a 6-fold excess of DisA relative to PcrA [27]. Furthermore, DisA neither compromises PriA-dependent replication initiation nor affects replication elongation using an in vitro reconstituted replication assay [13]. These results confirm that the inhibitory action exerted by DisA is specific for RuvAB.

### 2.7. DisA Suppresses the DNA-Dependent ATPase Activity of RuvAB

To understand the molecular basis of the inhibition of the RuvAB motor, we tested whether DisA inhibits RuvAB-mediated hydrolysis of ATP. As previously reported for RuvA*_Eco_* or RuvA [37,41,55], RuvA lacked any ATPase activity (K_cat_ of <0.1 min^−1^) in the presence or absence of ssDNA (Table 1). In the absence of DNA, RuvB or the RuvAB complex (15 nM) hydrolyzed ATP with a similar K_cat_ (~280 ± 7 min^−1^) (Table 1). Similarly, RuvB*_Eco_* can hydrolyze ATP in the absence of any DNA [37,55]. The DAC activity of DisA does not form ADP, the product that our assay is measuring [18].

In the presence of circular pGEM-3Zf(+) ssDNA (cssDNA, 10 μM in nt), the ATPase activity of RuvAB (1 RuvAB complex/666-nt) was stimulated by ~5-fold (*p* < 0.01), but it only increased ~2-fold (*p* < 0.05) in the presence of HJ-J3 DNA (10 μM in nt) (Figure 4A vs. Figure 4B, dark blue line, Table 1), as observed for RuvAB*_Eco_* [31]. In the presence of 10 mM Mg^2+^, cssDNA adopts secondary structures with single- (mimicking an unreplicated fork) and double-hairpin motifs (analogous to a HJ). It is likely that these structures mimic the preferred structures for RuvAB binding (Figure 4A vs. Figure 4B), as earlier reported for RuvAB*_Eco_* [56].

In the presence of cssDNA and limiting DisA concentrations (6 nM, ~2 DisAs/cssDNA molecule), RuvAB-mediated ATP hydrolysis was significantly reduced (by ~4-fold, *p* < 0.01) (Figure 4A, orange line, Table 1). Moreover, in the presence of nearly stoichiometric DisA concentrations (12 nM) with respect to RuvAB, the maximal rate of ATP hydrolysis was comparable to that of RuvAB in the absence of DNA (*p* > 0.1) (Figure 4A, green vs. dark blue line, Table 1). Similarly, in the presence of HJ-J3 DNA and DisA, the rate of RuvAB-mediated ATP hydrolysis was comparable to that of RuvAB in the absence of DNA (*p* > 0.1) (Figure 4B, brown vs. dark blue line, Table 1). It is likely that DisA interacts with and inhibits the ssDNA-dependent activity of RuvB (Figure 2A,B), but it does not affect the DNA-independent ATPase activity of RuvB (or of the RuvAB complex).

### 2.8. DisA Does Not Compete with RuvAB for the Binding to DNA

DisA might compete with RuvAB for DNA binding and inhibit its ssDNA-dependent ATPase activity. To test this hypothesis, DisA was replaced by DisA ΔC290. This mutant, which lacks the DNA-binding domain but retains its DAC activity [24], interacts with RuvB (Figure 2B). In the presence of ssDNA or HJ DNA, RuvAB-mediated ATP hydrolysis was inhibited by DisA ΔC290, to levels comparable to *wt* DisA (Figure 4A,B vs. Figure 4C,D, orange, green, and brown vs. dark blue line, Table 1). This suggests that DisA does not inhibit RuvAB-mediated ATPase activity due to competition for DNA binding. Since DisA neither inhibits the DNA-independent RuvB (or RuvAB) ATPase nor competes with RuvAB for binding to its DNA substrate, we assumed that the inhibition of the RuvAB ATPase by DisA is a genuine and specific activity of the latter. Furthermore, DisA does not affect the ATPase activity of the unrelated PcrA DNA helicase [27].

### 2.9. The DAC Activity of DisA Does Not Compromise RuvAB-Mediated ATP Hydrolysis

DisA might exhaust the ATP pool with its DAC activity, or the c-di-AMP produced might poison RuvAB ATPase activity. To examine this hypothesis, DisA was replaced by the DisA D77N variant, which cannot synthesize c-di-AMP [18,22]. In the presence of ssDNA or HJ-J3 DNA and DisA D77N, RuvAB-mediated ATP hydrolysis was inhibited to a comparable level to that when *wt* DisA was used (Figure 4A,B vs. Figure 4E,F, orange, green, and brown vs. dark blue line, Table 1). Therefore, neither the DAC activity of DisA with subsequent consumption of ATP, nor c-di-AMP, are involved in the reduction of RuvAB-mediated ATP hydrolysis. Altogether, these data suggest that DisA inhibits the ATPase activity of the RuvAB-ssDNA complex by its direct interaction with RuvB.

### 2.10. DisA Does Not Interact with RecU

With few exceptions, the RecU HJ resolvase constitutes, together with the RuvAB complex, the resolvasome in bacteria of the Firmicutes and Tenericutes phyla [41]. Genetic studies suggest that *disA* is epistatic to the *recA* or *recU* gene in response to DNA damage [13,23]. The RecA protein physically interacts with both DisA and RecU [18,57]. Upon cell envelope stress, the expression of both the *disA* and the *recU* genes is significantly increased [35,36], suggesting a temporal and functional link between them. Therefore, a functional interaction of DisA and RecU was explored. First, we analyzed whether RecU and DisA physically interact using the bacterial two-hybrid system. RecU interacts with itself, in agreement with the X-ray data that show that the protein is a dimer [42]. A stable interaction of RecU with DisA, however, was not observed (Figure 5A).

To analyze whether DisA modulates the mechanism of RecU-mediated HJ resolution, the RecU HJ resolvase was purified, and DNA binding was analyzed. In the presence of 1 mM Mg^2+^, RecU specifically binds HJ DNA, and in the presence of 10 mM Mg^2+^, RecU binds and cleaves the HJ structure when its cognate site is exposed [42,46]. Therefore, to perform DNA binding studies, we used 1 mM MgCl_2_, which it is not the optimal condition for DisA binding (see Appendix A, and Appendix A).

RecU binds HJ-J3 DNA with high affinity (K_Dapp_ of 0.6 ± 0.2 nM; Figure 5B, lanes 5–7), as described [46] and with ~30-fold higher affinity than DisA under this experimental condition. When HJ-J3 DNA was incubated with a fixed amount of RecU and increasing DisA concentrations, independently of the order of addition, protein-HJ DNA complexes became entrapped in the well (Figure 5B, lanes 9, 10, 12 and 13), so that the presence of a putative RecU-HJ-DisA complex remained poorly defined.

To further analyze a potential interaction, we made DNAse I protection footprint assays. Since DNase I requires 10 mM MgCl_2_ for its activity, but under this condition RecU binds and cleaves the HJ structure, the [γ^32^P]-HJ-J3 DNA-protein complexes were first pre-formed at 1 mM Mg^2+^ (5 min at 37 °C). Then, fixed DNase I and MgCl_2_ up to 10 mM were added (Figure 5C). RecU mainly interacts with the junction region, which is DNAse I resistant. Thus, a clear protected area was not observed. However, a defined band within the ssDNA junction was observed (Figure 5C, lanes 2 and 3, marked with an asterisk), whose position correlates with the expected cleavage of HJ DNA by RecU [46]. The footprint results are consistent with the observation that the RecU stalk region, by penetrating in the center of the HJ, distorts it so that the HJ adopts a square planar conformation with a central hole [36,48]. In the presence of fixed RecU and increasing DisA concentrations, the band corresponding to RecU-mediated cleavage of HJ DNA significantly decreased, and the extended footprint of DisA was observed (Figure 5C, lanes 6 and 7). It is likely that both proteins co-exist on HJ DNA, and that DisA either alters the positioning of RecU, the conformation of the HJ DNA, or both, thus affecting RecU-mediated HJ cleavage.

### 2.11. DisA Limits RecU-Mediated HJ Cleavage

The effect of DisA on RecU-mediated HJ cleavage was then analyzed. A large excess of DisA did not catalyze HJ cleavage (Figure 5D, lane 2). In the presence of 10 mM Mg^2+^, RecU cleaved the HJ-J3 to yield two nicked duplex products (Figure 5D, lane 3) [46,48]. When RecU was pre-incubated with HJ-J3 DNA (5 min 37 °C), the addition of a stoichiometric concentration of apo DisA only marginally affected RecU-mediated HJ-J3 cleavage (*p* > 0.1) (Figure 5D, lane 3 vs. 6). In sharp contrast, when apo DisA was pre-incubated with HJ-J3 DNA, and then RecU was added, HJ cleavage was strongly inhibited even at a molar DisA:RecU ratio of 0.2:1 (Figure 5D, lane 7), and it was abolished in the presence of sub-stochiometric DisA concentrations (Figure 5D, lanes 8 and 9). Since the inhibition exerted by DisA was dependent on the order of addition, we assumed that it was genuine. Here, ATP was not present in the reaction mixture, thus DisA-mediated c-di-AMP synthesis plays no role.

### 2.12. RuvAB- or RecU-Bound HJ-J3 DNA Blocks DisA-Mediated C-Di-AMP Synthesis

DisA catalyzes the synthesis of the essential c-di-AMP second messenger that regulates a wide variety of physiological functions and plays a central role in virulence in bacteria of the Firmicutes phylum [16]. DisA inhibits RuvAB- and RecU-mediated HJ processing (Figure 3D and Figure 5D), and in the presence of HJ DNA, DisA-mediated c-di-AMP synthesis is inhibited [22,23]. We tested whether RuvAB or RecU regulates the DisA DAC activity at limiting ATP concentrations (K_m_ 151 ± 1.4 μM) [27] to reduce the branch migrating activity of RuvAB that would disassemble the HJ DNA. Under the conditions used to detect radiolabeled [α-^32^P]-c-di-AMP, [α-^32^P]-ADP is poorly separated from the [α-^32^P]-ATP substrate; therefore, RuvAB-mediated ATP hydrolysis cannot be detected in our assay. In the presence of 100 μM ATP and 10 mM MgCl_2_, DisA converted 70–80% of the ATP substrate into product (c-di-AMP), and traces of the pppA-pA intermediate were detected (Figure 6A,B, lane 2), as described [22,23]. The addition of RuvAB or RecU (30 nM) marginally affected the DAC activity of DisA (*p* > 0.1) (Figure 6A,B, lane 7 vs. 2).

As expected [22,23], in the presence of a fixed HJ-J3 DNA concentration, DisA-mediated c-di-AMP synthesis was reduced by ~3-fold (*p* < 0.05) (Figure 6A,B, lanes 3 vs. 2). In the presence of HJ-J3 DNA and RuvAB or RecU, several outcomes can be expected when the DisA DAC activity is assayed. First, RuvAB (or RecU) may displace DisA by competing for binding or by unwinding or cleaving, respectively, the HJ DNA, leading to the recovery of the DAC activity of DisA. Second, RuvAB (or RecU) may stabilize or relocate DisA on the HJ-J3 structure, thus additively affecting DisA-mediated c-di-AMP synthesis. Third, DisA bound to HJ DNA may not interact with RuvB, and the DAC activity is inhibited as in the presence of only HJ DNA. We found that in the presence of HJ-J3 DNA and a limiting RuvAB concentration (RuvAB:DisA 0.3:1 molar ratio), the DAC activity of DisA was additively inhibited (~6-fold) (*p* < 0.01), suggesting that RuvAB can stabilize or relocate DisA on HJ-J3 DNA (see above). At RuvAB:DisA ratios approaching stoichiometry (0.6:1 and 1.2:1), RuvAB slightly reversed this negative effect, but the DisA DAC activity was still inhibited when compared with HJ DNA alone (Figure 6A, lane 3 vs. 5 and 6). To understand this mechanism of inhibition, the order of protein addition was varied. When DisA was pre-incubated with HJ-J3 DNA, and then RuvAB at a 1.2:1 RuvAB:DisA molar ratio was added, DisA-mediated c-di-AMP synthesis was strongly inhibited, suggesting that RuvAB stabilizes or relocates DisA on the HJ-J3 structure and this further impedes the DAC activity. However, if RuvAB was pre-incubated with the HJ-J3 DNA, the DAC activity of DisA was partially recovered at about stoichiometric concentrations (RuvAB:DisA 1.2:1 molar ratio) (Figure 6A, lane 9 vs. 3), perhaps because RuvAB translocates the HJ-J3 DNA. This suggests that there is a complex interplay between the three components (RuvAB, DisA, and HJ-J3 DNA). RuvAB bound to HJ DNA unwinds HJ structures, unless DisA is prebound to the HJ, and, on the other hand, DisA bound to HJ DNA suppresses its c-di-AMP synthesis in the presence of RuvAB.

Then, RuvAB was replaced by RecU. Increasing RecU concentrations in concert with HJ-J3 DNA synergistically inhibited the DAC activity of DisA (Figure 6B, lanes 4–6 vs. 7). When DisA was pre-incubated with HJ-J3 DNA, and then RecU was added, DisA-mediated c-di-AMP synthesis was strongly inhibited, suggesting that RecU may not displace DisA from the HJ-J3 structure (Figure 6B, lane 8 vs. 3). However, if RecU was pre-incubated with the HJ-J3 DNA, the DAC activity of DisA was partially recovered (Figure 6B, lane 9 vs. 3). These results suggest that RuvAB or RecU may stabilize and relocate DisA on the HJ-J3 DNA, but the RuvAB-HJ DNA or RecU-HJ DNA complexes may process the DNA substrate and indirectly reduce the inhibition exerted by HJ DNA.

## 3. Discussion

In response to a replication stress, a stalled fork can be remodeled, but the function(s) that process(es) a stalled fork and the molecular basis of its regulation are poorly characterized in bacteria other than those of the γ-Proteobacteria class. Genetic analysis showed that when the single genome of an inert mature *B. subtilis* spore is exposed to ionizing radiation and then the spores are revived under unperturbed conditions, RecA, RecG, RuvAB, RecU, and DisA are required for survival, but neither RecQ-like (RecS and RecQ) remodelers nor the end-resection functions (AddAB and RecJ) are involved in spore revival [12,13].

Cytological studies have shown that DisA scans the chromosome and pauses at RecA bound to lesion-containing gaps at stalled replication forks rather than at DSBs during sporulation [17,18]. Dynamic DisA also pauses in unperturbed exponentially growing ∆*recU* cells but not in the exponentially growing *wt* cells [23]. In the presence of MMS- or the UV mimetic 4-nitroquinoline 1-oxide (4NQO)-induced lesions, RecA filamented on the ssDNA gaps triggers the DNA damage response and stops DisA scanning, perhaps loading it at stalled or reversed forks [18]. DisA bound to branched intermediates reduces c-di-AMP synthesis to levels comparable to that in the absence of DisA, in otherwise *wt* cells [21]. Low c-di-AMP levels indirectly inhibit cell proliferation (see the introduction).

Taking these data into account, we hypothesized that DisA could be providing, together with fork remodelers, a mechanism to cope with replicative stress and could protect stalled forks. Previously, it has been shown that during replication stress, DisA limits the activities of RecA and RecG [18,27]. Here, we show that DisA interacts with RuvB and inhibits the DNA-dependent ATPase activity of the RuvAB complex (Figure 2A,B and Figure 4). RuvAB fails to reverse a stalled fork at protein concentrations that efficiently regress a reversed fork, and DisA cannot stimulate the conversion of a stalled into a reversed fork (Figure 3). It is likely that during a replication stress, RecG (or RecA) converts a stalled fork into a reversed fork [27]. The reversed fork may be further processed by the RuvAB-RecU resolvasome to yield a one-ended DSB and a nicked duplex, as reported in *E. coli* [10,28,29]. However, if this occurs, these physiological reversed forks intermediates would become a pathological structure, as during the early stages of spore revival (see the introduction). We show here that DisA limits the processing of a reversed fork by the RuvAB-RecU resolvasome (Figure 3 and Figure 5), and accordingly, it partially reduces chromosomal breakage of unperturbed growing cells (Figure 2F and Figure 7).

From the results presented here and in previous reports, we propose that, in response to a 4NQO- or MMS-induced insult, the replisome disengages from the DNA [58], exposing a stalled fork to be remodeled, with SsbA (counterpart of SSB*_Eco_*) coating the ssDNA lesion-containing gap region [5,59]. Second, with the help of mediators and modulators, RecA forms a nucleoprotein filament [60]. This nucleoprotein filament induces the SOS response (increasing RecA and RuvAB levels), and a membrane stress increases DisA and RecU levels. Third, dynamic DisA, which scans for its cognate target site [17], interacts with the RecA nucleoprotein filament and with its cognate site (a stalled or reversed fork) and pauses there, also reducing RecA dynamics [18]. Static DisA bound to a branched intermediate (or the static RecA nucleoprotein filament) might prevent the degradation of the nascent strands of the reversed fork (Figure 2F). Fourth, RecG (or RecA) bound to a stalled fork could convert it into a reversed fork (see above). Fifth, RuvB interacts with DisA and RecU at reversed forks [36,57]. Sixth, DisA bound to a reversed fork partially suppresses c-di-AMP synthesis, and RuvAB or RecU, upon binding to a DisA-HJ DNA complex, synergistically block c-di-AMP synthesis (Figure 6). Seventh, low c-di-AMP levels increase the production of (p)ppGpp, which in turn directly inhibits the DNA primase and indirectly inhibits cell proliferation [25,26], to avoid the uncoupling of the cell cycle. Eighth, in the absence of DisA, RuvAB bound to HJ DNA could branch migrate it until the RecU cognate site is exposed, and thereby assists RecU to cleave the HJ structure [41]. Ninth, DisA, acting as an “emergency brake”, limits/delays RuvAB-mediated fork remodeling and RecU-mediated HJ cleavage to stabilize a stalled or reversed fork, to prevent nascent strand degradation and genome instability. In other words, DisA bound to a reversed fork protects the extruded nascent strands from an unscheduled cleavage by the RuvAB-RecU resolvasome until the lesion is circumvented (Figure 2, Figure 4 and Figure 7a,b). When the 3′-end of the nascent leading-strand is elongated, using it as a template the nascent lagging-strand, RuvAB (or RecG) could remodel the HJ structure back to a fork structure, and this process is also modulated by DisA (Figure 7c,d). Finally, the offending lesion is removed from duplex DNA, the fork reconstituted, RuvAB and RecU disassembled from the DNA, and replication restarted. DisA recovers its dynamic behavior and catalyzes c-di-AMP synthesis in order to reactivate cell proliferation (Figure 7).

In summary, our present findings reveal that DisA, recruited by RecA (see the introduction), may protect a stalled or reversed fork from degradation as proposed for the eukaryotic mediator BRCA2 and/or Rad51 itself [9]. DisA, as a guardian of genome integrity, provides a quality control to prevent a physiological reversed fork from becoming a pathological one during spore revival by interacting with and/or limiting RuvAB and RecU (this work) or RecG activities [27]. Moreover, a variation in the order of protein recruitment might generate different outcomes to increase survival under different conditions (e.g., sporulating cells and reviving spores vs. exponentially growing cells). The presence of DisA in non-spore-forming bacteria (e.g., *Mycobacterium tuberculosis*) also provides a conceptual framework for future studies exploring the DisA regulatory balance to overcome replicative stress and its broader role in genome stability in bacteria.

## 4. Materials and Methods

### 4.1. Bacterial Strain and Plasmids

*E. coli* BL21(DE3) cells bearing pLysS and pET-derived plasmids were used for protein overexpression. XL1-Blue cells were used for cloning and plasmid amplification, and BTH101 cells bearing plasmids pUT18 and pUT18C (to generate fusions at the N- and C-termini of the T18 domain, respectively), pKNT25 and pKT25 (to generate fusions at the N- and C-termini of the T25 domain, respectively), and its derivatives were used for bacterial two-hybrid analyses. *E. coli* plasmids pCB875, pCB568, pCB632, pCB1080, and pCB1081 were used to over-express and purify the DisA, RecU, RuvA, DisA D77N, and DisA ΔC290 proteins, respectively [21,24,41,46]. *B. subtilis* BG214 cells bearing the pCB737-borne *ruvB* gene were used to over-express and purify RuvB [61].

### 4.2. Protein–Protein Interaction Assays

In vivo protein–protein interaction was assayed using the bacterial adenylate cyclase-based two-hybrid (BACTH) technique as described [24]. The plasmid-borne DisA (or DisA ΔC290 (which lacks the C-terminal 70 residues)) fusions to the T18 or T25 catalytic domain of the *Bordetella* adenylate cyclase, either at the N- (DisA-T18 and DisA-T25) or C-terminus (T18-DisA or T25-DisA), were pairwise co-transformed into the reporter BTH101 strain with plasmid-borne RuvA, RuvB, or RecU fusions, also to the T18 or T25 catalytic domain, either at the N- or the C-terminus. The empty vectors or the pKT25-Zip and pUT18C-Zip vectors were co-transformed into the reporter strain as negative and positive controls, respectively. Serial dilutions were spotted onto LB plates supplemented with ampicillin, kanamycin, streptomycin, 0.5 mM IPTG, and 10% X-Gal. The plates were then incubated at 25 °C for 3–4 days and photographed. Each co-transformation was performed at least in triplicate and a representative result is shown.

### 4.3. Pulse-Field Gel Electrophoresis

Cultures of *B. subtilis* BG214 (*trpCE metA*5 *amyE1 ytsJ*1 *rsbV*37 *xre*1 *xkd*A1 *att*^SPß^ *att*^ICE*Bs*1^) and its isogenic derivatives lacking DisA (BG1221, Δ*disA*) or RecU (BG855, Δ*recU*) [21,61] were grown at 37 °C in LB medium to an OD_560_ of 0.6. Then, cells were treated or not with 10 mM MMS for 20 min. A volume of culture corresponding to ~2 × 10^8^ cells was then centrifuged (14,000 rpm, 5 min at 4 °C) and washed twice with 1.5 mL of TEN buffer (50 mM Tris-HCl pH 8, 50 mM EDTA, 100 mM NaCl). Finally, the cell pellet was resuspended in 0.1 mL of TEN buffer containing 2 μg of lysozyme, mixed with an equal volume of 2% Certified Low Melt Agarose (Bio-Rad, Hercules, CA, USA) brought to 55 °C, dispensed into wells of disposable molds (Bio-Rad), and solidified. The agarose plugs were incubated for 20 h at 50 °C in NDS buffer (10 mM Tris-HCl pH 8.0, 0.4 M EDTA pH 8.0, 1% *N*-lauryl sarcosine) with 0.35 mg/mL proteinase K. After completion of the incubation, the lysis buffer was replaced with TE buffer and the plugs were stored at 4 °C until used. A CHEF-DR II pulsed-field gel electrophoresis system (Bio-Rad) was used to resolve the DNA. Running conditions were 20 h, 7 °C, 5 V/cm, with a pulse time of 5–80 s. The 1% agarose gel was stained with ethidium bromide and photographed. The densitometric analysis of the lanes was performed using the Image Lab software (Bio-Rad). The percentage of chromosomal fragmentation was calculated as the signal between 610 and 48 Kb in the lane, divided by the combined signal of the lane plus well. The experiment was performed three times, and results are plotted relative to the value obtained with the *wt* in the untreated condition. *t*-tests were applied to analyze the statistical significance of the data.

### 4.4. DNA Substrates

The nucleotide (nt) sequences of the oligonucleotides used are indicated in the 5′→3′polarity: J3-1, CGCAAGCGACAGGAACCTCGAGAAGCTTCCGGTAGCAGCCT GAGCGGTGGTTGAATTCCTCGAGGTTCCTGTCGCTTGCG; J3-2, CGCAGCGACAGG AACCTCGAGGAATTCAACCACCGCTCAACTCAACTGCAGTCTGACTCGAGGTTC CTGTCGCTTGCG; J3-3, CGCAAGCGACAGGAACCTCGAGTCTAGACTGCAGTTGA GTCCTTGCTAGGACGGATCCCTCAGGTTCCTGTCGCTTGCG; J3-4, CGCAAGCGAC AGGAACCTCGAGGGATCCGTCCTAGCAAGGGGCTGCTACCGGAAGCTTCTCGA GGTTCCTGTCGCTTGCG; J3-5, CGCAAGCG ACAGGAACCTCGAGTCTAGACTGCA GTTGAGTTGAGCGGTGGTTGAATTCCTCGAGT TCCTGTCGCTTGCG; J170, CTAGA GACGCTGCCGAATTCTGGCTTGGATCTGATGCTGTCTAGAGGCCTCCACTATGA AATCGCTGCA; J173, CCGGGCTGCAGAGCTCATAGATCGATAGTCTCTAGACAGC ATCAGATCCAAGCCAGAATTCGGCAGCGTCT; J345, GCGATTTCATAGTGGAGGC CTCTAGACAGCACGCCGTTGAATGGGCGGATGCTAATTACTATC TC; J346, GAGA TAGTAATTAGCATCCGCCCATTCAACGGCGTGCTGTCTAGAGACTATCGATCTAT GAGCTCTGCAGC; 170, AGACGCTGCCGAATTCTGGCTTGGATCTGATGCTGTCTA GAGGCCTCCACTATGAAATCG; 173, AGCTCATAGATCGATAGTCTCTAGACAGC ATCAGATCCAAGCCAGAATTCGGCAGCGTCT; 171, CGATTTCATAGTGGAGGCCT CTAGACAGCA; 172, TGCTGTCTAGAGACTATCGATCTATGAGCT; 171-15, CGATTT CATAGTGGA and 172-15, ATCGATCTATGAGCT. The dsDNA was assembled by annealing J3-2 and J3-5 (80-bp); fixed HJ (HJ-J3 DNA) by annealing J3-1, J3-2, J3-3, and J3-4; mobile HJ (HJ-J4 DNA) by annealing J170, J173, J345, and J346; flayed by annealing 170 and 173; forked-Lag by annealing 170, 173, 171, and 172-15; and forked-Lead by annealing 170, 173, 171-15, and 172. The ssDNA concentrations were measured using the extinction coefficient of 1.54 × 10^−4^ M^−1^ cm^−1^ at 260 nm, and the concentrations of DNA substrates are expressed as moles of DNA molecules or moles of nucleotides as indicated. Annealing was performed by mixing the appropriate oligonucleotides in 50 mM phosphate buffer pH 7.5, heating for 5 min at 100 °C, and then slowly cooling. The annealed products were gel purified as described previously, dialyzed against buffer A (50 mM Tris-HCl (pH 7.0), 5% glycerol, 1 mM DTT), and stored at 4 °C [42].

### 4.5. Protein Purification

The RecU, RuvA, RuvB, DisA, DisA D77N, and DisA ΔC290 proteins were purified as described [18,23,24,41,46]. The proteins were >95% pure based on staining after SDS-PAGE, and partial proteolysis and MALDI-TOF analysis. RuvA was free of the RuvA*_Eco_* protein. The molar extinction coefficients for DisA, RecU, RuvA, and RuvB were calculated as 22,350, 27,850, 11,900, and 13,400 M^−1^ cm^−1^ at 280 nm, as described [57]. The concentrations of RecU, RuvA, RuvB, and DisA (and its DAC active site [DisA D77N] or DNA-binding domain [DisA ΔC290] mutant) are expressed as moles of dimers, tetramers, hexamers, and octamers, respectively.

### 4.6. ATPase Activity and c-di-AMP Synthesis

The ATP hydrolysis activity of the RuvAB protein was assayed via an ATP/NADH-coupled spectrophotometric enzymatic assay. Assays were done in buffer B (50 mM Tris-HCl pH 7.5, 80 mM NaCl, 10 mM Mg(CH_3_COO)_2_, 50 μg/mL bovine serum albumin (BSA), 1 mM DTT, 5% glycerol) containing 5 mM ATP and an ATP regeneration system (620 μM NADH, 100 U/mL of lactate dehydrogenase, 500 U/mL pyruvate kinase, and 2.5 mM phosphoenolpyruvate) for 30 min at 37 °C, as described.

The order of addition of DNA effectors (circular 3199-nt pGEM3 Zf (+) ssDNA or HJ DNA [10 μM in nt]) and purified proteins is indicated in the text. The data obtained from the rate of NADH absorbance decrease at 340 nm is proportional to the rate of ADP production, and it is plotted as a function of time. As reported [59], the rate of ATP hydrolysis (K_cat_) was derived from the slope of the linear part of the curves. *t*-tests were applied to analyze the statistical significance of the data.

c-di-AMP formation was analyzed using thin-layer chromatography (TLC) and [α-^32^P]-ATP as described [22,23]. Reactions were performed at 37 °C using a range of protein concentrations, in buffer D (50 mM Tris-HCl pH 7.5, 50 mM NaCl, 1 mM DTT, 0.5 μg/mL BSA, 0.1% Triton, 5% glycerol) containing 100 μM ATP (at a ratio of 1:2000 [α^32^P]-ATP:ATP) and the indicated MgCl_2_ and HJ DNA concentration. After 30 min of incubation, the reaction was chelated by adding 50 mM EDTA, and 2 μL of each reaction were spotted onto 20 × 20 cm TLC polyethyleneimine cellulose plates and run for about 2 h in a TLC chamber containing running buffer E [1:1 (*v*/*v*) 1.5 M KH_2_PO_4_ (pH 3.6) and 70% ammonium sulfate]. Dried TLC plates were analyzed by phosphorimaging and spots were quantified using ImageJ (NIH). *t*-tests were applied to analyze the statistical significance of the data.

### 4.7. DNA Binding, HJ Branch Migration, and Cleavage Assays

DNA binding was assayed by EMSA using different [γ^32^P]-labelled DNA substrates (0.2 nM in molecules). The radiolabeled strand is indicated with an asterisk. The binding was performed in buffer C (50 mM Tris-HCl pH 7.5, 50 mM NaCl, 1 mM DTT, 0.05 mg/mL BSA, 5% glycerol) containing 1 or 10 mM MgCl_2_ or 5 mM EDTA and when indicated a nucleotide cofactor. Reactions were incubated for 15 min at 37 °C. Prior to loading, 0.2% glutaraldehyde was added to stabilize the complexes. Protein-DNA complexes were separated using 6% PAGE in TAE buffer and visualized by autoradiography. Autoradiography films were scanned, and the ImageJ software (NIH, Bethesda, MD, USA) was used to determine the signal from each band and obtain the apparent binding constant (K_Dapp_) values at the protein concentration that gives 50% of DNA-protein complexes. *t*-tests were applied to analyze the statistical significance of the data.

The reaction conditions for DNase I footprint experiments were the same as for EMSA. DNase I treatment was performed as described [62]. The samples were resuspended in loading buffer [80% (*v*/*v*) formamide, 10 mM NaOH, 1 mM EDTA, 0.1 % (*v*/*v*) bromophenol blue, and 0.1% (*v*/*v*) xylene cyanol], separated in 15% denaturing PAGE (dPAGE), and autoradiographed. For the size control marker, ladders obtained with the chemical sequencing reaction (G + A) on the same DNA fragments were used.

Cleavage of HJ-J3 DNA (labeled on arm 1) at the indicated concentrations of RecU was assayed for 30 min at 37 °C in buffer C containing 10 mM MgCl_2_. When indicated, increasing concentrations of DisA were added. After deproteinization by the addition of one-fifth volume of stop mix (5% SDS, 100 mM EDTA, 5 mg/mL proteinase K) and further incubation for 10 min at 37 °C, the products of the cleavage were analyzed by 15% denaturing PAGE and autoradiography. *t*-tests were applied to analyze the statistical significance of the data.

In a standard branch migration assay, [γ^32^P]-labelled HJ-J4 DNA (0.1 nM in molecules) was incubated with RuvAB in buffer C containing 10 mM MgCl_2_ and 5 mM ATP for 30 min at 37 °C. When indicated, increasing concentrations of DisA were added. Reactions were terminated by adding one-fifth volume of stop mix (5% SDS, 100 mM EDTA, 5 mg/mL proteinase K) and further incubation for 10 min at 37 °C to deproteinize the sample. Unwound products were analyzed by 6% PAGE in TAE buffer and phosphorimaging. Signals of substrates and products were quantified with the ImageJ software (NIH). *t*-tests were applied to analyze the statistical significance of the data.

## Figures and Tables

**Figure 1 ijms-22-11323-f001:**
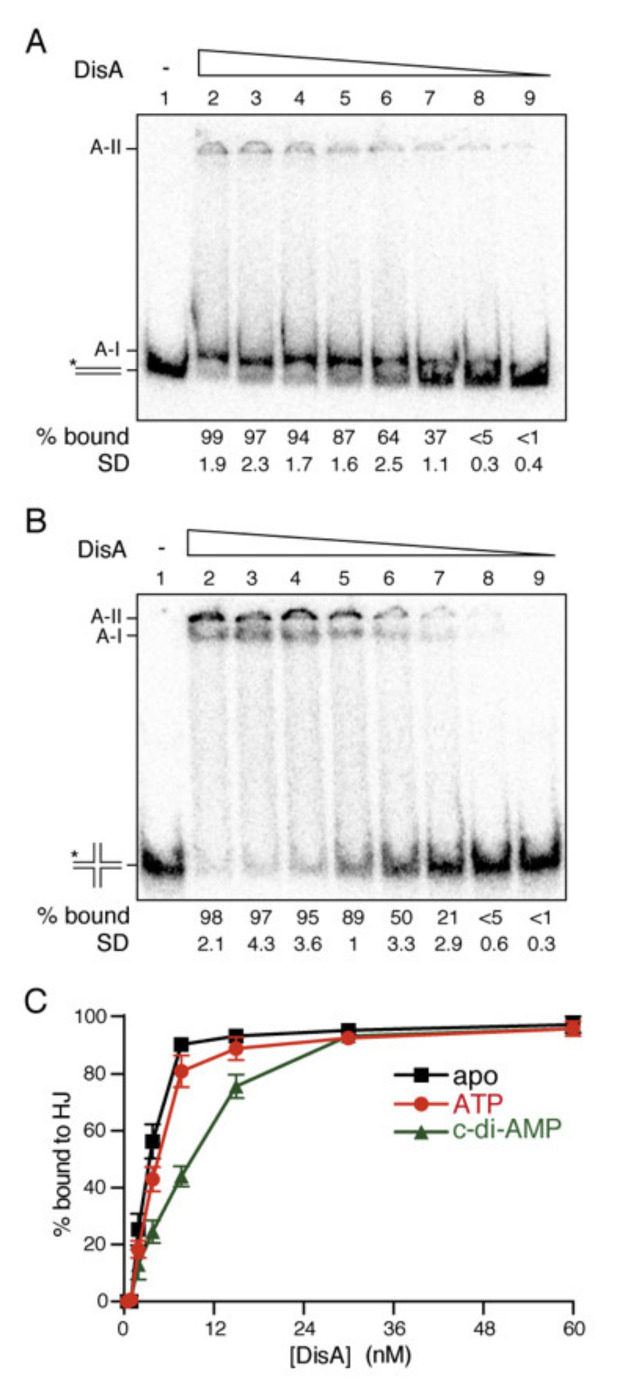
DisA binding to DNA. [γ-^32^P] dsDNA (**A**) or [γ-^32^P] HJ DNA (**B**) was incubated with increasing DisA concentrations (doubling from 0.45 to 60 nM) in buffer C containing 10 mM MgCl_2_, and after fixation, complexes were detected by EMSA. Experiments were repeated at least 3 times. A representative gel and below the mean % of DNA bound and its SD are shown. A-I denotes complexes entering the gel and A-II denotes complexes retained in the well. (**C**) ATP or c-di-AMP barely affects DisA binding to HJ DNA. [γ-^32^P] HJ DNA was incubated with increasing DisA concentrations (15 min at 37 °C) in buffer C containing 10 mM MgCl_2_ or 10 mM MgCl_2_ and 500 µM ATP or c-di-AMP (15 min at 37 °C). The complexes, detected by EMSA, were quantified. Results are shown as the mean ± SEM of >3 independent experiments.

**Figure 2 ijms-22-11323-f002:**
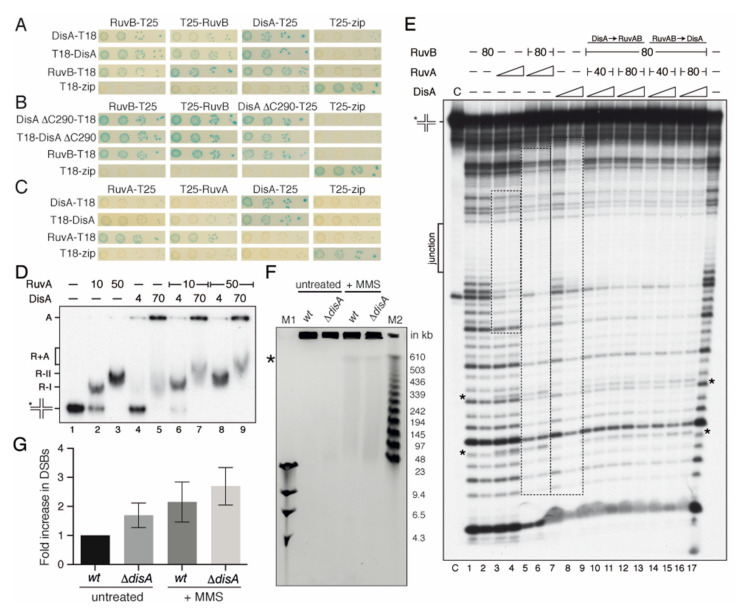
DisA interacts with RuvB and may contribute to fork protection. (**A**–**C**) Bacterial two-hybrid interaction assays were done, co-transforming the pair of plasmids expressing full-length DisA, DisA ΔC290, RuvB, or RuvA fused at the N- or C-terminus, to either the T18 or T25 domain of the *Bordetella* adenylate cyclase. A positive interaction was observed by the appearance of blue color. Experiments were repeated at least 3 times, and representative photographs are shown. In the plates, four serial dilutions of transformed cells were spotted. (**D**) DisA binds RuvA-HJ DNA complexes. [γ-^32^P] HJ DNA was incubated with the indicated RuvA and/or DisA concentrations for 15 min in buffer C containing 1 mM MgCl_2_ at 37 °C. The protein-HJ DNA complexes were separated by 6% native PAGE. -, no protein added. RuvA bound to HJ forms two types of complex (R-I and R-II), DisA-HJ mainly one type of complex (A) and RuvA-HJ-DisA (R+A) complexes. (**E**) DNase I footprint analyses. [γ-^32^P] HJ DNA was pre-incubated with a fixed amount of RuvB (80 nM), and increasing concentrations of RuvA (40 and 80 nM) or DisA (60 and 120 nM) in buffer C containing 5 mM ATPγS and 10 mM MgCl_2_ (for 15 min at 37 °C). Then, the second protein (DisA or RuvAB) was added, and the reaction incubated (15 min at 37 °C). Finally, DNAse I was added. C, the HJ DNA control without DNase I, and in lanes 1 and 17 with DNase I treatment. The regions protected by the individual proteins or by both proteins are marked with rectangles. The order of protein addition is indicated at the top. The position of the ssDNA crossover is indicated as ‘*junction*’. The hypersensitive sites characteristic of RuvA binding are highlighted by an asterisk. Experiments were repeated at least 3 times, and a representative gel is shown. (**F**) PFGE of *wt* and Δ*disA* cells after treatment or not with 10 mM MMS for 20 min before the preparation of plugs. *, DSB compression zone (above 600 kb), smaller DSBs are detected as a smear. Markers used are Lambda PFG ladder (M1) and Lambda DNA *Hin*dIII digest (M2). (**G**) Quantification of the chromosomal fragmentation. The increase in DSBs was plotted relative to the *wt* untreated condition, which is given a value of 1. Results are the mean plus the SD of three independent experiments.

**Figure 3 ijms-22-11323-f003:**
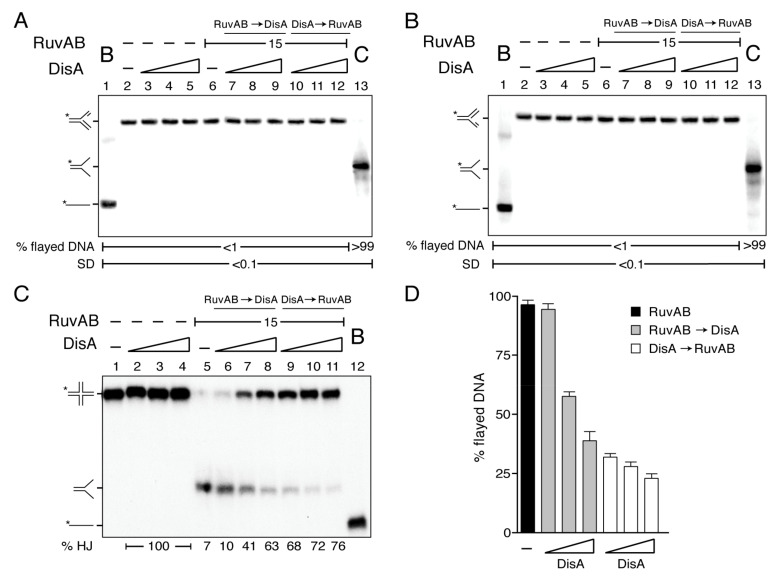
DisA action on RuvAB translocase activity. (**A**,**B**) RuvAB fails to convert a gapped stalled fork into a reversed one. [γ-^32^P] forked-Lead (**A**) or [γ-^32^P] forked-Lag DNA (**B**) was pre-incubated with increasing DisA concentrations (doubling from 12–48 nM) or a fixed amount of RuvAB (15 nM) in buffer C containing 10 mM MgCl_2_ (15 min at 37 °C). Then, the second protein (variable DisA [RuvAB → DisA] or a constant amount of RuvAB [DisA → RuvAB]) and 5 mM ATP were added, and the reaction was further incubated (15 min at 37 °C). The reaction was stopped, deproteinized, and separated by 6% native PAGE. -, no protein added; B, boiled forked substrate (lane 1); *, radiolabelled strand; C, expected products of the helicase activity loaded as running position controls. Representative gels are shown. (**C**) DisA inhibits RuvAB-mediated fork restoration. [γ-^32^P] HJ-J4 DNA was pre-incubated with increasing DisA concentrations or a fixed amount of RuvAB in buffer C containing 10 mM MgCl_2_ (15 min at 37 °C). Then, the second protein (variable DisA [RuvAB → DisA] or a constant amount of RuvAB [DisA → RuvAB]) and 5 mM ATP were added, and the reaction was further incubated (15 min at 37 °C). The reaction was stopped, deproteinized, and separated by 6% native PAGE. -, no protein added; B, boiled substrate. The percentage of remaining substrate (HJ) is indicated at the bottom. (**D**) The relative amount of flayed DNA produced in three independent experiments as the one shown in (**C**) was quantified. Values are represented as the mean ± SEM.

**Figure 4 ijms-22-11323-f004:**
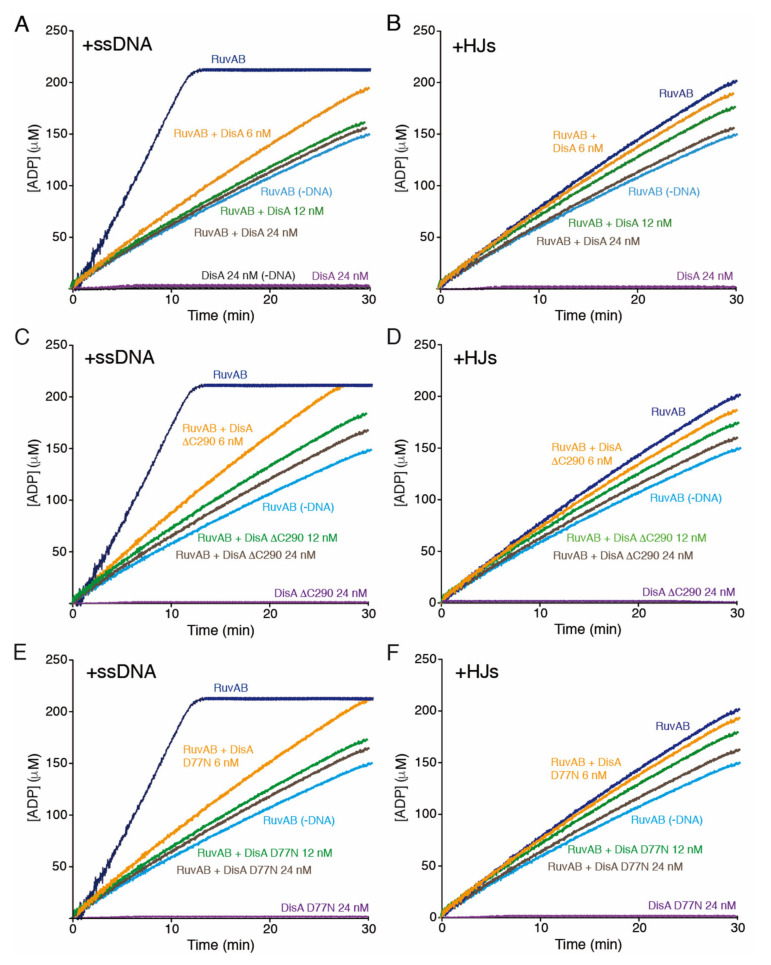
DisA inhibits RuvAB-mediated ATP hydrolysis. (**A**,**B**) The ATPase activity of the RuvAB complex (15 nM) in buffer B containing 5 mM ATP was measured during 30 min with circular 3199-nt ssDNA (in **A**) or HJ DNA (in **B**) as the effector. As indicated, DisA (6, 12 and 24 nM) was added, or DNA was omitted. (**C**–**F**) DisA inhibits RuvAB-mediated ATP hydrolysis by a direct interaction. Circular 3199-nt ssDNA (10 μM in nt) (**C**,**E**) or HJ DNA (**D**,**F**) was incubated with RuvAB (15 nM) and DisA ΔC290 (**C**,**D**) or DisA D77N (**E**,**F**) (6 to 24 nM) in buffer B containing 5 mM ATP, and the ATPase activity was measured (30 min at 37 °C). All reactions were repeated three or more times with similar results, representative graphs are shown here, and the mean K_cat_ ± SEM are listed in Table 1.

**Figure 5 ijms-22-11323-f005:**
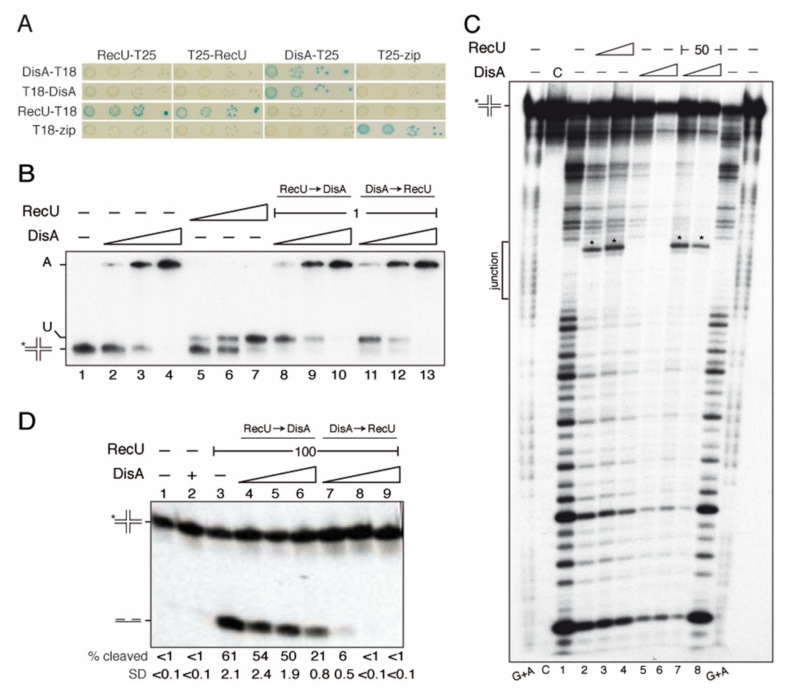
DisA does not interact with RecU but inhibits RecU-mediated HJ cleavage. (**A**) DisA interaction with RecU was not observed by the bacterial two-hybrid interaction assay. (**B**) [γ-^32^P] HJ DNA was pre-incubated with increasing DisA concentrations (doubling from 12–48 nM) or RecU (doubling from 0.25–1 nM) for 15 min in buffer C containing 1 mM MgCl_2_ (5 min at 37 °C). Then, the second protein (variable DisA or a constant amount of RecU) was added and the reaction was incubated (15 min at 37 °C). The protein-HJ DNA complexes were separated by 6% native PAGE. -, no protein added. The RecU-HJ (U) and DisA-HJ (A) complexes are shown. The order of protein addition is indicated at the top. (**C**) Autoradiogram showing a footprint analysis of the binding of RecU and DisA to HJ DNA. [γ-^32^P] HJ DNA was pre-incubated with increasing concentrations of RecU (50 and 100 nM) or DisA (60 to 120 nM) (5 min at 37 °C) in buffer C containing 1 mM MgCl_2_. Then, the second protein (RecU 50 nM) was added, and the reaction incubated (15 min at 37 °C). After that, DNase I and MgCl_2_ up to 10 mM were added. C, no DNase I was added. The position of the ssDNA crossover is indicated as ‘*junction*’ and RecU-mediated cleavage of the [γ-^32^P]-labelled arm is indicated by an *. The G + A marker is indicated. Experiments were repeated at least 3 times, and a representative gel is shown. (**D**) [γ-^32^P] HJ DNA labelled on arm 1 (denoted by *) was pre-incubated with DisA (doubling from 24–96 nM) or RecU (100 nM) (5 min at 37 °C) in buffer C containing 1 mM MgCl_2_. Then, the second protein (variable DisA [RecU → DisA] or a constant amount of RecU [DisA → RecU]), and MgCl_2_ up to 10 mM were added, and the reaction was incubated (15 min at 37 °C). The reaction was stopped, deproteinized, and analyzed by 15% denaturing PAGE, no protein added. The relative amount of cleaved DNA in three independent experiments was quantified as described, and a representative gel, the mean % of cleaved DNA, and its SD are shown.

**Figure 6 ijms-22-11323-f006:**
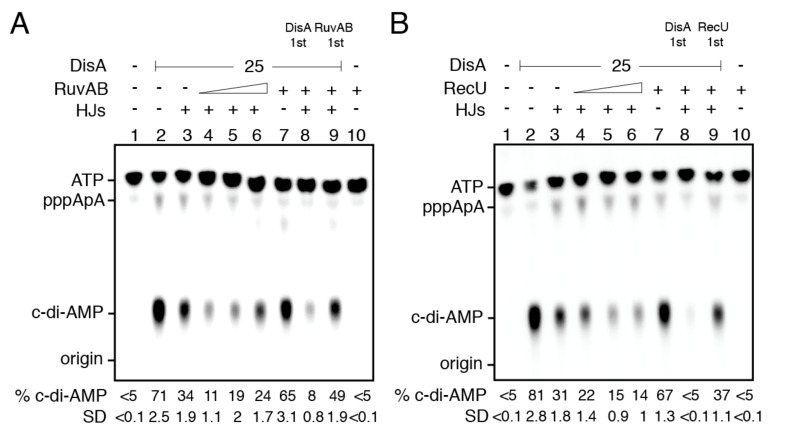
HJ DNA and RuvAB or RecU inhibits DisA-mediated c-di-AMP synthesis. (**A**,**B**) A fixed DisA concentration was incubated alone (lane 2), with HJ DNA (125 nM) (lane 3), increasing RuvAB (**A**) or RecU (**B**) concentrations (7, 15, or 30 nM) and HJ DNA (lanes 4–6), or with RuvAB (**A**) or RecU (**B**) (30 nM, lane 7) in buffer D containing 100 μM ATP (at a ratio of 1:2000 [α^32^P]-ATP:ATP) and 10 mM MgCl_2_ (30 min at 37 °C). In lane 8, DisA was pre-incubated with HJ DNA (5 min at 37 °C) and then a fixed concentration of RuvAB (**A**) or RecU (**B**) and ATP were added, and the reaction incubated (30 min at 37 °C). In lane 9, RuvAB or RecU was pre-incubated with HJ DNA and then DisA and ATP were added, and the reaction was incubated (30 min at 37 °C). In lane 10, a control reaction with only RuvAB or RecU (30 nM) is shown. The substrates and products were separated by TLC, and the spots quantified. The position of ATP, linear pppApA, c-di-AMP, and the origin are indicated. At least three independent experiments were performed, a representative plate, and the mean % of c-di-AMP produced and its SD are shown.

**Figure 7 ijms-22-11323-f007:**
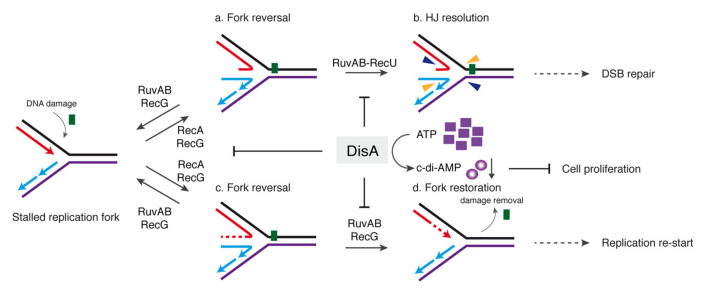
Proposed DisA mode of action in the presence of RecU and RuvAB. An unrepaired DNA lesion on the leading-strand template (green square) causes blockage of replication fork movement. Fork reversal by the RecA or RecG remodeler may form an HJ DNA structure and this process is inhibited by DisA. In the upper panel, DisA suppresses fork reversal and HJ resolution by the RuvAB-RecU resolvasome (**a**,**b**), avoiding the formation of a one-ended DSB. In the lower panel, DNA synthesis extends the regressed fork to convert it into an HJ-like structure, and fork restoration is catalyzed either by RuvAB or RecG (**c**,**d**), followed by damage removal by specific repair mechanisms. This process is also downregulated by DisA. DisA bound to HJ DNA decreases c-di-AMP synthesis, indirectly increasing (p)ppGpp synthesis and inhibiting cell proliferation.

**Table 1 ijms-22-11323-t001:** RuvAB rates of ATP hydrolysis.

Protein ± DNA	K_cat_ (min^−1^) ^a^
RuvA (15 nM), no DNA	<0.1
RuvB (15 nM), no DNA	283 ± 5
RuvAB (15 nM), no DNA	280 ± 7
RuvAB + HJ DNA	660 ± 13
RuvAB + HJ DNA + 6 nM DisA ^b^	645 ± 11
RuvAB + HJ DNA + 12 nM DisA ^c^	554 ± 9
RuvAB + HJ DNA + 24 nM DisA ^d^	321 ± 11
RuvAB + HJ DNA + 6 nM DisA D77N ^b^	651 ± 11
RuvAB + HJ DNA + 12 nM DisA D77N ^c^	578 ± 7
RuvAB + HJ DNA + 24 nM DisA D77N ^d^	354 ± 9
RuvAB + HJ DNA + 6 nM DisA ΔC290 ^b^	640 ± 15
RuvAB + HJ DNA + 12 nM DisA ΔC290 ^c^	547 ± 11
RuvAB + HJ DNA + 24 nM DisA ΔC290 ^d^	351 ± 10
RuvAB + ssDNA	1253 ± 19
RuvAB + ssDNA + 6 nM DisA ^b^	405 ± 8
RuvAB + ssDNA + 12 nM DisA ^c^	329 ± 4
RuvAB + ssDNA + 24 nM DisA ^d^	301 ± 5
RuvAB + ssDNA + 6 nM DisA D77N ^b^	433 ± 13
RuvAB + ssDNA + 12 nM DisA D77N ^c^	350 ± 9
RuvAB + ssDNA + 24 nM DisA D77N ^d^	330 ± 8
RuvAB + ssDNA + 6 nM DisA ΔC290 ^b^	461 ± 14
RuvAB + ssDNA + 12 nM DisA ΔC290 ^c^	375 ± 11
RuvAB + ssDNA + 24 nM DisA ΔC290 ^d^	341 ± 9

^a^ ATP hydrolysis was measured as indicated in the materials and methods. The protein(s) was pre-incubated with ssDNA or HJ DNA (10 μM in nt). The stoichiometry of DisA with DNA is indicated (1 DisA octamer/1600- ^b^, 800- ^c^, 400-nt ^d^). The kinetic parameters for RuvAB (1 protein/667-nt) were derived from the data presented in Figure 4. The average rate of ATP hydrolysis was obtained from more than three independent experiments, and it is shown as the mean ± SEM.

## Data Availability

The datasets analyzed or generated during the study are avalaible upon request.

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
