# Peer review of "DisA Restrains the Processing and Cleavage of Reversed Replication Forks by the RuvAB-RecU Resolvasome"

_ijms, 2021, doi:10.3390/ijms222111323_

Round 1
Reviewer 1 Report
Gandara et al show in this work how the scanning protein DisA interferes with the role of RuavAB-RecU resolvasome in replicative fork reversal. They show that DisA can bind both dsDNA and holiday junction (HJ) substrates but result in different mobility complex visualized by EMSAs. Using two-hybrid interaction assay, as well as biochemical evidences, they demonstrate the specific interaction of DisA with RuvB but not with RuvA in the context of HJ substrates. Authors show that translocation activity of RuvAB is impaired on HJ but not in gapped stalled forks and that ATPase activity of RuvAB is reduced in the presence of DisA. They show evidences that DisA is not interacting with RecU nor in the context of HJs. Looking at the enzymatic activity of DisA, c-di-AMP synthesis, they found that is inhibited by RuvAB/HJ or RecU/HJ, or just by the presence of the HJs.
The research is appropriately designed and the manuscript well written in a logical order. Authors present strong experimental evidences supporting their claims mainly that Dis A affects how RuvAB-RecU protein machinery deals with fork reversal by a direct interaction with RuvB.
Minor points:
Figure 1 legend: indicate what AI and AII are
Page 155 line 155: spell out HhH
Page 5 line 166: correct typo in ‘catalyzeds’
Page 5 lane 174: as written the sentence is misleading. What is not affecting the DAC activity? The GA crosslinking?
Page 6 lane 221: What is the rationale to investigate the C-terminal deletion of DisA and not the N-terminal?
Page 10 lanes 376-381 are repetition from 370-375
Author Response
Reviewer 1 (the queries are reproduced [denoted in italic] and the answer in bold)
Gandara et al show in this work how the scanning protein DisA interferes with the role of RuvAB-RecU resolvasome in replicative fork reversal. They show that DisA can bind both dsDNA and holiday junction (HJ) substrates but result in different mobility complex visualized by EMSAs. Using two-hybrid interaction assay, as well as biochemical evidences, they demonstrate the specific interaction of DisA with RuvB but not with RuvA in the context of HJ substrates. Authors show that translocation activity of RuvAB is impaired on HJ but not in gapped stalled forks and that ATPase activity of RuvAB is reduced in the presence of DisA. They show evidences that DisA is not interacting with RecU nor in the context of HJs. Looking at the enzymatic activity of DisA, c-di-AMP synthesis, they found that is inhibited by RuvAB/HJ or RecU/HJ, or just by the presence of the HJs. The research is appropriately designed and the manuscript well written in a logical order. Authors present strong experimental evidences supporting their claims mainly that Dis A affects how RuvAB-RecU protein machinery deals with fork reversal by a direct interaction with RuvB.
Thank you for your enthusiasm about our biochemical characterization of the interplay of B. subtilis DisA with RuvAB and RecU.
Minor points:
Figure 1 legend: indicate what AI and AII are
Done. The information is now included in the legend to the figure
Page 155 line 155: spell out HhH
Done. HhH stands for helix-hairpin helix.
Page 5 line 166: correct typo in ‘catalyzeds’
Done
Page 5 lane 174: as written the sentence is misleading. What is not affecting the DAC activity? The GA crosslinking?
The sentence was rephrased.
Page 6 lane 221: What is the rationale to investigate the C-terminal deletion of DisA and not the N-terminal?
The DisA DNA-binding domain (HhH domains), which is structurally similar to RuvA, lies in the C-ter region of the protein. Both DisA and RuvA interact with RuvB. Thus, to test whether RuvB interacts with DisA through this domain we used a C-ter deletion mutant of DisA. The piece of information is now included at the beginning of the paragraph to answer the query.
Page 10 lanes 376-381 are repetition from 370-375
Thank you for spotting the repeated text, it was generated when we tried to fit the figures in the corresponding page.
Reviewer 2 Report
This manuscript by Gándara et al. analyzes the effect of DisA on the actions of the RuvAB and RecU proteins in processing DNA structures that model intermediates in the repair of stalled DNA replication forks. They build on the group's previous findings where they identify RecA, RuvAB, RecU, RecG and DisA as genes required for spore survival in B. subtilis. Their previous work has focused on the effect of DisA on RecA and, more recently, RecG. In this last work published in Cells in 2021 the authors show how DisA limits the action of RecG on its substrates without a direct interaction and how RecG and DisA do not compete for the binding to the different structures and DisA activity is inhibited when it is mobility is reduced. The present work shows a similar effect of DisA on RuvAB and RecU. The data are solid, the experiments are well performed and correctly presented, and the conclusions are supported by the results. However, the results in the manuscript do not show a significant advance compared to their previous publications since the authors just show that DisA has a similar effect on different components of the same pathway. There is no significant mechanistical or conceptual advance to support the publication of the work in International Journal of Molecular Sciences. The individual analysis of the effect of DisA on each of these proteins does not answer the more relevant outstanding questions. The authors suggest that the effect on RuvAB is mediated by a direct interaction but they do not map the interaction domain. Also, they propose that the effect on RecU depends on DNA binding but they do not use the DNA binding mutant to prove this point. There is no information on whether all these functions are necessary in vivo. Actually, the model proposed by the authors in Figure 7 is reproduces half of the last figure in their paper in Cells reinforcing the idea that the impact of this work is limited. Thus, this manuscript basically reproduces similar data previously published by the authors and only extends their observations without providing a significant advance.
Author Response
Reviewer 2 (the queries are reproduced [denoted in italic] and the answer in bold)
This manuscript by Gándara et al. analyzes the effect of DisA on the actions of the RuvAB and RecU proteins in processing DNA structures that model intermediates in the repair of stalled DNA replication forks. They build on the group's previous findings where they identify RecA, RuvAB, RecU, RecG and DisA as genes required for spore survival in B. subtilis. Their previous work has focused on the effect of DisA on RecA and, more recently, RecG. In this last work published in Cells in 2021 the authors show how DisA limits the action of RecG on its substrates without a direct interaction and how RecG and DisA do not compete for the binding to the different structures and DisA activity is inhibited when it is mobility is reduced. The present work shows a similar effect of DisA on RuvAB and RecU. The data are solid, the experiments are well performed and correctly presented, and the conclusions are supported by the results. However, the results in the manuscript do not show a significant advance compared to their previous publications since the authors just show that DisA has a similar effect on different components of the same pathway. There is no significant mechanistical or conceptual advance to support the publication of the work in International Journal of Molecular Sciences.
Thank you for your enthusiasm about our biochemical characterization of the interplay of B. subtilis DisA with RuvAB and RecU. We would like to highlight that although both, RecG and RuvAB-RecU, contribute to the remodeling of stalled forks, they do not act in the same pathway in E. coli. Furthermore, in S. aureus ruvAB is an essential gene, and in B. subtilis ruvAB recG cells are not viable, suggesting that in Firmicutes they perform different activities in the cell. Therefore, the effect of DisA over these two different recombination processing machineries could be different and we thought it deserved a separated analysis. The study of the effect of DisA over RecG has been published in Cells this year, and here we report the effects of DisA on the RuvAB-RecU resolvasome. We show that RuvAB cannot reverse stalled fork, as same researchers also show for E. coli RuvAB (e.g. Marians´s and McGlynn groups).
The individual analysis of the effect of DisA on each of these proteins does not answer the more relevant outstanding questions. The authors suggest that the effect on RuvAB is mediated by a direct interaction but they do not map the interaction domain. Also, they propose that the effect on RecU depends on DNA binding but they do not use the DNA binding mutant to prove this point.
We agree that we have not mapped the DisA domain required for interaction with RuvB. Monomeric DisA consists of an N-terminal DAC domain and a C-terminal DNA binding domain joined by a spine like linker. We show that deletion of the HhH domain does not affect the DisA-RuvB interaction, suggesting that the interaction maps in the N-terminal DAC domain or in the link between both domains.
The DisA mutant lacking the DNA binding domain was not used with RecU. However, by changing the order of protein addition we observe a competition effect. From these results it can be inferred that the DisA mutant lacking the DNA binding domain will not inhibit RecU.
There is no information on whether all these functions are necessary in vivo.
We are confused with this query. Previously, we have shown that: i) ruvAB or recU inactivation compromises cell viability and chromosomal segregation in unperturbed exponentially growing cells (Carrasco et al 2004; Sanchez et al 2005), ii) inactivation of disA renders cells sensitive and of ruvAB or recU very sensitive to methyl methane sulfonate or the UV mimetic 4-nitroquiniline (Sanchez et al 2005, 2007, Torres et al 2017, Gándara et al 2017); and iii) the disA gene is epistatic to ruvAB or recU in response to ionizing radiation damage. All these references are properly cited in the text.
Actually, the model proposed by the authors in Figure 7 is reproduces half of the last figure in their paper in Cells reinforcing the idea that the impact of this work is limited.
The model depicted in Figure 7 contains additional information with respect to the one published in Cells. When the interplay between RecG and DisA was published, we were not aware that RuvAB could not catalyze fork reversal. From the data of Figure 3 we have learned that RuvAB cannot remodel a stalled fork, but it can catalyze fork restoration. This result is not trivial, because it is under strong controversy between researcher working with E. coli. Since the absence of RuvAB-mediated fork reversal is also documented in a distantly genetic bacterium (B. subtilis) here, this suggests that it is not a genuine activity associated with the purified protein. Moreover, the functional interplay between DisA and RuvAB is depicted in Figure 7, and was still unknown in our previous paper in Cells.
Thus, this manuscript basically reproduces similar data previously published by the authors and only extends their observations without providing a significant advance.
We are confused, please remember that the ruvAB or recU gene product is essential in S. aureus, and that ruvAB is synthetically lethal in the recG or recD2 context in B. subtilis.
As far as we are aware this is the first report characterizing the activity of the B. subtilis RuvAB translocase on replication and recombination intermediates, other than the Holliday junction, as well as the interplay between DisA with RuvAB and RecU.
In summary, we previously characterized the interplay of DisA with RecG (Torres et al 2021). Here, we have used a very similar experimental design used in the analysis of the interplay between DisA and RecG to be able, in the future, to compare the role of DisA with RuvAB and RecU side-by-side with DisA and RecG.
Reviewer 3 Report
This review manuscript written by Gándara et al described biochemical characterization of the B. subtilis DisA protein to investigate the function of this protein in the stalled fork repair process. RecA interacts with and facilitates DisA pausing at the branched intermediates. DisA has c-di-AMP cyclase, but its enzymatic activity is suppressed by binding to stalled fork structure. The RuvAB-RecU resolvasome branch-migrates and resolves formed Holliday junctions (HJ). DisA interacts with RuvB, a component of the resolvasome, and binds Holliday junctions (HJ), and reduces the DNA-dependent ATPase activity of RuvAB, suggesting that the HJ-bound DisA inhibits branch migration by RuvAB and resolution by RecU. This inhibition does not appear if the resolvasome by RuvAB and RecU is pre-formed. The pre-formed resolvasome strongly inhibits DisA-mediated synthesis of c-di-AMP, and indirectly blocks cell proliferation. Based on these data, the authors propose that DisA can provide some time to the repair the damage by stopping RuvAB-mediated fork remodeling and RecU-mediated HJ to preserve genome integrity. I think this work is interesting for many readers of the journal and encourage publication. I have several questions and comments, which should be solved clearly to improve the manuscript.
- This work is biochemical characterization of DisA related to RuvAB, RecU, and HJ DNA, and therefore, purity of these proteins is very important. The SDS-PAGE image of these purified proteins should be shown in a Suppl figure.
- The EMSA assay in Figure 1 needs more information. A-I and A-II should be indicated in the legend. Actually, what is A-I? Why the mobility of A-I is so different between HJ and dsDNA? The procedure to calculated the apparent KD should be described in the Materials and Methods. The signal on the wells of the gel cannot be identified whether they are from specific binding of DisA. An agarose gel electrophoresis with 1-2% should be work to detect a specific band instead of 6% PAGE. Otherwise, the KD values calculated in this study do not have specific meading. RuvA-HJ complex is clearly shown in Fig. 2D. The band identified in the well is not likely dependent on the protein size. How about pI value of DisA?. In addition, the probe DNAs shown in Suppl Fig 1 should have not only the calculated table, but also the gel images. It is not easy to understand the obvious different binding by different Mg2+ concentrations, 1mM and 10 mM. Binding was observed even with 5 mM EDTA (without any help of Mg2+). Some more explanation is required.
- Figure 2. A-C. What are the four spots for each combination of the bacterial 2H assay? F-G. It looks that the most right lane is not the size markers but the MMS-treated genome DNA from ΔdisA
- Figure 6. The enzymatic assay for c-di-AMP synthesis activity. One problem is that c-di-AMP and ADP could not be separated by the method shown in this figure. The positive controls for c-di-AMP and ADP by themselves should be loaded with the investigated reaction condition for this experiment. Actually, why the product spot was increased with increasing amount of RuvAB in A (lanes 4-6), but not RecU in B (lanes 4-6). The spot may be the mixture of c-di-AMP and ADP, because RuvAB has ATPase activity.
- This work showed the model, in which DisA works for genome stability of B. sultilis and also other bacteria possessing DisA (Figure 7). The authors propose that the stalled fork-bound DisA stops Fork remodeling and cell proliferating. These things are reasonable from the results of this work. However, one thing that is not clear is how DisA is removed from the stalled fork and how repair process starts. Some explanation should be added how the authors think about it, actually what the switch of the next step is.
Author Response
Reviewer 3 (the queries are reproduced [denoted in italic] and the answer in bold)
This review manuscript written by Gándara et al described biochemical characterization of the B. subtilis DisA protein to investigate the function of this protein in the stalled fork repair process. RecA interacts with and facilitates DisA pausing at the branched intermediates. DisA has c-di-AMP cyclase, but its enzymatic activity is suppressed by binding to stalled fork structure. The RuvAB-RecU resolvasome branch-migrates and resolves formed Holliday junctions (HJ). DisA interacts with RuvB, a component of the resolvasome, and binds Holliday junctions (HJ), and reduces the DNA-dependent ATPase activity of RuvAB, suggesting that the HJ-bound DisA inhibits branch migration by RuvAB and resolution by RecU. This inhibition does not appear if the resolvasome by RuvAB and RecU is pre-formed. The pre-formed resolvasome strongly inhibits DisA-mediated synthesis of c-di-AMP, and indirectly blocks cell proliferation. Based on these data, the authors propose that DisA can provide some time to the repair the damage by stopping RuvAB-mediated fork remodeling and RecU-mediated HJ to preserve genome integrity. I think this work is interesting for many readers of the journal and encourage publication. I have several questions and comments, which should be solved clearly to improve the manuscript.
Thank you for your enthusiasm about our biochemical characterization of the interplay of B. subtilis DisA with RuvAB and RecU.
- This work is biochemical characterization of DisA related to RuvAB, RecU, and HJ DNA, and therefore, purity of these proteins is very important. The SDS-PAGE image of these purified proteins should be shown in a Suppl figure.
We believe that to show SDS-gel using sobre-expression systems does not provide much information that our sentence in the main text “The proteins were > 95% pure based on staining after SDS-PAGE, and partial proteolysis and MALDI-TOF analysis”. To answer the query we provide an SDS-gel in a separate document attached to this letter for the reviewer’s evaluation, and in accordance with the policy of the Journal any information not included in the manuscript is available upon request.
2. The EMSA assay in Figure 1 needs more information. A-I and A-II should be indicated in the legend. Actually, what is A-I? Why the mobility of A-I is so different between HJ and dsDNA? The procedure to calculated the apparent KD should be described in the Materials and Methods.
The information about the meaning of A-I and A-II is now included in the legend of Figure 1. As highlighted by the referee the mobility of gel entering complexes is very different between HJ DNA and dsDNA. We propose that DisA binds dsDNA in a different manner than to HJ DNA, and thus the complexes show a different mobility. This is consistent with the biochemical result that the interaction of DisA with HJ DNA (and in less extent with ssDNA) suppresses its DAC activity, but the interaction with duplex DNA does not. Cytological data showed that DisA transiently interacts with dsDNA in its scanning mode, whereas once DNA replication is arrested by the presence of DNA damage, DisA pauses (Bejerano-Sagie et al 2006). Recently we have shown that DisA pausing requires RecA but not AddAB or RecJ, suggesting that DisA pauses at stalled or reversed forks. The procedure to calculate the apparent KD is now described in Materials and Methods (Page 16, lanes 776-779)
3. The signal on the wells of the gel cannot be identified whether they are from specific binding of DisA. An agarose gel electrophoresis with 1-2% should be work to detect a specific band instead of 6% PAGE. Otherwise, the KD values calculated in this study do not have specific meaning. RuvA-HJ complex is clearly shown in Fig. 2D. The band identified in the well is not likely dependent on the protein size. How about pI value of DisA?. In addition, the probe DNAs shown in Suppl Fig 1 should have not only the calculated table, but also the gel images.
We respectfully disagree with the reviewer. A Kd value suggests how specific or not specific is DNA binding. We cannot compare the mobility of RuvA-HJ DNA complexes with the DisA-HJ complexes. In fact, we expected a lower mobility for DisA-HJ DNA complexes. RuvA is a tetramer that only interacts with one side of a HJ molecule. In contrast, DisA is an octamer that interacts with HJ molecules in a complex form. We do not believe that mobility shift assays with agarose gels will give a better resolution than the DNase footprinting assays that we performed. A similar experimental design was used for E. coli RuvAB complexes. These studies show an interaction of DisA with at least two arms of the HJ DNA. We believe that gel images of the data presented in Suppl Fig 1 will provide little insight in the manuscript. We include the original gels in a separate document attached to this letter for the reviewer’s evaluation, and in accordance with the policy of the Journal any information not included in the manuscript is available upon request.
4. It is not easy to understand the obvious different binding by different Mg2+ concentrations, 1mM and 10 mM. Binding was observed even with 5 mM EDTA (without any help of Mg2+). Some more explanation is required.
We are confused. DisA binding to DNA requires Mg2+, and it is favored by 10 mM Mg2+. At about KDapp concentrations of DisA (5 nM) about 90% of the protein is bound to HJ DNA in the presence of 10 mM Mg2+, and no binding was observed in the presence of 5 mM EDTA (Figure S1B). We have pointed out to this result in the main text to make this statement more-clear. The text in page 4 now reads: Similarly, binding to DNA was best at high than at low MgCl2 concentrations or in the presence of EDTA (Figure S1A-B). It is likely that in the presence of physiological Mg2+ concentrations (10 mM), DisA forms a stable macro-complex with HJ-J3 DNA that is not entering in a gel and suppresses its DAC activity. DisA, however, forms only a transient interaction with duplex DNA, that was captured in our EMSA assays because of the use of glutaraldehyde. Thus, the presence of dsDNA does not affect DisA-mediated DAC activity (Figure S2).
5. Figure 2. A-C. What are the four spots for each combination of the bacterial 2H assay? F-G. It looks that the most right lane is not the size markers but the MMS-treated genome DNA from ΔdisA
As stated in Materials and Methods: “Serial dilutions were spotted onto LB plates supplemented with ampicillin, kanamycin, streptomycin, 0.5 mM IPTG and 10% X-Gal.”. This is now explained in the figure legend. Regarding the second point, there was a displacement in the figure that suggested that the most-right lane was not a marker. We provide now a new figure 2 with this error corrected. In Figure 2F, the markers are labelled, and explained in the figure legend.
6. Figure 6. The enzymatic assay for c-di-AMP synthesis activity. One problem is that c-di-AMP and ADP could not be separated by the method shown in this figure. The positive controls for c-di-AMP and ADP by themselves should be loaded with the investigated reaction condition for this experiment. Actually, why the product spot was increased with increasing amount of RuvAB in A (lanes 4-6), but not RecU in B (lanes 4-6). The spot may be the mixture of c-di-AMP and ADP, because RuvAB has ATPase activity.
We feel sorry if the message was not clear. The assay is optimized for detecting c-di-AMP, and under this condition we do separate c-di-AMP from ADP and ATP, but we cannot separate ATP from ADP. DisA and RuvAB have a different Km for ATP (published reports). As stated in the text, with the ATP concentration used the branch migrating activity of RuvAB is significantly reduced, otherwise the HJ-DNA should be remodeled. Concerning the query “why the product spot was increased with increasing amount of RuvAB in A (lanes 4-6), but not RecU in B (lanes 4-6)”, this is explained in the text. It is likely that by increasing the RuvAB concentration the HJ DNA is remodeled and the relative concentration of HJ DNA is reduced, and indirectly the inhibition of c-di-AMP synthesis is partially released. This is not observed with RecU because we are using a HJ with no cognate site for RecU mediated cleavage, therefore the relative HJ concentration remains constant
7. This work showed the model, in which DisA works for genome stability of B. sultilis and also other bacteria possessing DisA (Figure 7). The authors propose that the stalled fork-bound DisA stops Fork remodeling and cell proliferating. These things are reasonable from the results of this work. However, one thing that is not clear is how DisA is removed from the stalled fork and how repair process starts. Some explanation should be added how the authors think about it, actually what the switch of the next step is.
We proposed in the Discussion section that when the damaged DNA is removed and replication restarts, there is no ssDNA region at which RecA could bind to load DisA and no HJ intermediates to which DisA, RuvAB or RecU could bind. Thus, DisA will recover its dynamic behavior. There could be also some proteins that may displace DisA from HJ DNA facilitating this recovery. We show here that RuvAB and RecU can do it, if they are first bound to the DNA (figure 6, RuvAB 1st or RecU1st conditions). This possibility is now highlighted in the last part of the discussion section.
Round 2
Reviewer 2 Report
As the authors state in their response "the effect of DisA over these two different recombination processing machineries could be different" to what they already described for RecG and this merits the analysis. However, their results show that it is not and thus, although this work offers a new analysis of the effect of DisA on RuvAB and RecU in B. subtilis, the conclusions are not so relevant. Similarly, the authors show for the first time that B. subtilis RuvAB acts on replication and recombination intermediates though E. coli RuvAB has been shown to act on these structures. The authors show different activities in this context but provide no further information as to why this might be.
In my opinion the manuscript still lacks relevance for publication in International Journal of Molecular Sciences. Mapping the interaction domain of DisA with RuvB would allow to test the relevance of this interaction in vivo, as I suggested. Showing the inferred conclusion that the DisA DNA binding mutant does not inhibit RecU is a simple experiment that could have been performed in this time. Also, comparing side by side the effects of DisA with RecG to those with RuvAB and RecU could provide more interesting insights.
Beyond the fact that DisA inhibits all these recombination processing machines using similar mechanisms, it would be interesting to determine if it has a preferential effect on one or the other. Especially in light of the model the authors propose where all these proteins work in the same pathway and could compete for DisA. Also DisA mutants lacking specific interactions would provide the relevance for these studies.
Regarding the model I think that using the same figure already published in a different paper only with a partial replacement or update is, at least, shady. If the authors want to highlight their findings they should use a new model/figure centered on their findings.